



# LoopStructural 1.0: Time aware geological modelling

Lachlan Grose[1], Laurent Ailleres[1], Gautier Laurent[2] and Mark Jessell[3]

[1]School of Earth Atmosphere and Environment, Monash University, Melbourne 3800, Australia
[2]Université d'Orléans, CNRS, BRGM, ISTO, UMR 7327, Orleans France.
[3] Mineral Exploration Cooperative Research Centre, School of Earth Sciences, UWA, Perth, Australia

*Correspondence to*: Lachlan Grose (lachlan.grose@monash.edu)

**Abstract.** In this contribution we introduce *LoopStructural*, a new open source 3D geological modelling python package (www.github.com/Loop3d/LoopStructural). *LoopStructural* provides a generic API for 3D geological modelling applications harnessing the core python scientific libraries *pandas*, *numpy* and *scipy*. Six different interpolation algorithms, including 3

discrete interpolators and 3 polynomial trend interpolators, can be used from the same model design. This means that different interpolation algorithms can be mixed and matched within a geological model allowing for different geological objects e.g. different conformable foliations, fault surfaces, unconformities to be modelled using different algorithms. Geological features are incorporated into the model using a time-aware approach, where the most recent features are modelled first and used to constrain the geometries of the older features. For example, we use a fault frame for characterising the geometry of the fault

surface and apply each fault sequentially to the faulted surfaces. In this contribution we use *LoopStructural* to produce synthetic proof of concepts models and a 86x52km model of the Flinders ranges in South Australia using *map2loop*.

## 1. Introduction

Understanding and characterising the geometry and interaction between geological features in the subsurface is an important stage in resource identification and management. A surface or combination of surfaces can be used to represent the subsurface

geometry of geological features or structural elements within 3D geological models (Caumon et al., 2009). There are two main approaches for representing surfaces in 3D geological models 1) where the surface is represented by directly triangulating control points defining the surface geometry or 2) where the surface is extracted as an iso-value or level set of an implicit function (Wellmann and Caumon, 2018). Explicit surface representation is usually time consuming and requires significant subjective user input because surfaces are usually sculpted to the modellers conceptual idea in a similar way to drawing

polylines in geographical information systems or using computer aided design software. Implicit surface representation involves approximating an unknown function that represents the distance to a geological surface. The implicit function can be queried anywhere throughout the model for the value or gradient of the function. The implicit function is fitted to observations that represent the distance to a geological surface (for stratigraphic horizons this may be the cumulative thickness) or the gradient of the function (on contact or off contact) observations. The topological relationship between different geological

features, e.g. horizons, faults interactions, intrusions and unconformities are incorporated using multiple implicit functions for



different components of the model. Implicit surface representation removes the need to generate surfaces and allows for the geological features to be represented directly by the implicit function value.

All implicit surface modelling techniques involve finding a combination of weighted basis functions that fit the geological
observations. There are two main approaches used for implicit surface modelling: 1) data supported approaches where the basis functions are estimated at the data points (Calcagno et al., 2008a; Cowan et al., 2003; Gonçalves et al., 2017; Hillier et al., 2014; Lajaunie et al., 1997) and 2) discrete interpolation where the basis functions are located on a predefined support (Caumon et al., 2013; Frank et al., 2007; Irakarama et al., 2018; Renaudeau et al., 2019).The algorithms are often linked to commercial software e.g. Leapfrog[1], 3D GeoModeller[2] and Gocad-SKUA[3]. These packages will usually only provide one
algorithm for interpolation making it difficult to compare different interpolation schemes. The algorithms are also usually black box algorithms with limited ability to change algorithm parameters, with no understanding of how the algorithm is implemented. A recent open source python library, Gempy (De La Varga et al., 2019), implements the implicit interpolation algorithm used in 3D *GeoModeller* using *Theano* a high performance computational library.

In this contribution we introduce the open-source *LoopStructural*; a 3D geological modelling python library based on the incremental contributions of Laurent et al., 2016; Grose et al., 2017,2018 and 2019. *LoopStructural* is a new geological modelling engine developed within the Loop[4] consortium (Ailleres et al., 2018). The core modelling library within *LoopStructural* depends on *scipy* (Virtanen et al., 2020), *numpy* (Van Der Walt et al., 2011) and *pandas* (pandas development team, 2020)*, the* core scientific python libraries. A visualisation module uses *LavaVu* (Kaluza et al., 2020), a minimal OpenGL
visualisation package allowing for models to be visualised within a *Jupyter* notebook environment. *LoopStructural* has been written using an object-oriented program design with class structures designed to allow for powerful inheritance and modularity. The design of *LoopStructural* allows development and research into geological modelling methods to be easily performed without having to rewrite boilerplate code for interpolation algorithms, visualisation and model interaction. *LoopStructural* is a modelling package allowing for multiple stratigraphic groups, faults, folds and unconformities to be
represented using implicit surfaces. Different interpolation algorithms can be used for interpolating these surfaces with the ability to mix and match interpolation algorithms depending on the surfaces type being modelled. *LoopStructural* has native implementation of discrete implicit modelling using a piecewise linear interpolation on a tetrahedral mesh (Caumon et al., 2013; Frank et al., 2007; Mallet, 2014, 2002), finite difference interpolation on a Cartesian grid (Irakarama et al., 2018; Renaudeau et al., 2018), fold interpolation using tetrahedral meshes (Laurent et al., 2016), and an interface to a generalised
Radial Basis Interpolation (Hillier et al., 2014).

---

[1] https://www.seequent.com/products-solutions/leapfrog-software/
[2] https://www.intrepid-geophysics.com/product/geomodeller/
[3] https://www.pdgm.com/products/skua-gocad/
[4] https://www.loop3d.org

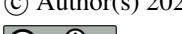



This paper begins with a background analysis of 3D modelling methods and the algorithms used in implicit modelling, with an overview of the mathematical and geological backgrounds used in our implementation. A detailed overview of the specifics of the implementation can be found on (loop3d.github.io/LoopStructural). To demonstrate the versatility of *LoopStructural*

and to provide a user guide we include three case studies in this paper with corresponding Jupyter notebooks. The first case study is a synthetic example interpolating two planar surfaces where the height of one surface has been perturbed to simulate uncertainty in the surface location. In this example we use the *LoopStructural* API to compare three different interpolation codes and investigate the parameters and how they are affected by noise. The second example is a synthetic refolded type 3 interference pattern (Laurent et al., 2016), where we apply the time-aware discrete fold interpolation method describe by

Laurent et al., 2016 for modelling the refolded folds. In the fourth and final case study *LoopStructural* is applied to a real dataset from the Flinders Ranges in South Australia, where the dataset has been prepared using the pre-processing module of the *Loop* workflow *map2loop* (Jessell et al., in prep).

## 2.   Materials and methods

A 3D geological model can be represented by a collection of surfaces representing geological features (e.g. fault surfaces,

stratigraphic horizons, axial surfaces of folds, unconformities) (Wellmann and Caumon, 2018). There are two main tasks for a 3D modelling software package:

- the creation of the surfaces from geological observations and knowledge, this is known as *interpolation;*
- the incorporation of geological concepts into the surface description e.g. faulted surfaces should show displacement and unconformities should be a boundary between units

In *LoopStructural* surfaces are implicitly represented by an isovalue of one or more volumetric scalar fields (Calcagno et al., 2008a; Caumon et al., 2013; Cowan et al., 2003; Frank et al., 2007; Gonçalves et al., 2017; Hillier et al., 2014; Jessell, 1981; De La Varga et al., 2019; Lajaunie et al., 1997; Mallet, 2002, 2014; Manchuk and Deutsch, 2019; Maxelon et al., 2009; Moyen et al., 2004; Renaudeau et al., 2019; Yang et al., 2019). The geological rules are managed by adding the geological event (folding event, one fault, another fault, an unconformity) structural parameters in a time aware approach, where the most recent

event is added first and the constraints are added backwards in time. Complex geological features such as folds and faults are integrated into *LoopStructural* by building a structural frame around the principal structural directions of the feature being modelled. Using these structural frames geological rules can be integrated into the modelling workflows – e.g. fault kinematics can be added to the faulted feature because the fault geometry is known before interpolating the faulted feature or fold overprinting relationships can be incorporated using multiple structural frames (Laurent et al., 2016).






### 2.1. Implicit surface modelling

Implicit surface modelling involves the representation of the geometry of a geological feature using a function $f(x, y, z)$ where the value of the function is the same along the observation of the surface. There are two possible ways of framing this question, the first approach uses the scalar field value as a distance from a reference horizon e.g. the location of the horizon for a single

surface would be the 0 value of the scalar field. Using this approach, which we will call the signed distance approach, the same implicit function can represent conformable horizons where the value of each horizon is the cumulative thickness from the base of the series (Caumon et al., 2013; Hillier et al., 2014; Jessell, 1981; Manchuk and Deutsch, 2019; Wellmann and Caumon, 2018). The second approach, can be named the potential field approach, does not specify the value of the scalar field. The scalar field approach only defines the scalar field to have the same value for specific interfaces, such as contacts between

geological units and fault traces (Calcagno et al., 2008a; De La Varga et al., 2019). As with the signed distance field, the potential field can represent conformable horizons - where the value of the implicit function evaluated on the input observations can be used to infer the potential field value for these horizons.

These implicit functions have no known analytical solution which means that they need to be approximated from the

observations that are provided. The implicit function is represented by a weighted combination of basis functions:

$$f(x, y, z) = \sum_{i=0}^{N} w_i \cdot \varphi_i(x, y, z)$$

Where $N$ is the number of basis functions, $w$ are the weights and $\varphi$ are the basis functions. There are two approaches for approximating the implicit function: the first approach uses a discrete formulation for the interpolation where $N$ is defined by some sort of mesh (Caumon et al., 2013; Frank et al., 2007; Mallet, 1992; Moyen et al., 2004); and the second approach uses

data supported basis functions where $N$ is defined by the number of data points (Calcagno et al., 2008a; Cowan et al., 2003; Gonçalves et al., 2017; Hillier et al., 2014; De La Varga et al., 2019; Lajaunie et al., 1997).

### 2.1.1. Input data

Geological observations that are directly incorporated into 3D modelling can generally be divided into two categories:

observations that describe the orientation of a geological feature (on contact and off contact)and observations that describe the location within a geological feature (cumulative thickness for conformable stratigraphic horizons, or location of fault surface) In the context of a geological map, location observations may be the trace of a geological surface on the geological map, or a single point observation at an outcrop or from a borehole. Orientation observations generally record a geometrical property of the surface - e.g. a vector that is tangential to the plane or the vector that is normal to the plane (black and grey arrows in Figure

120  1).




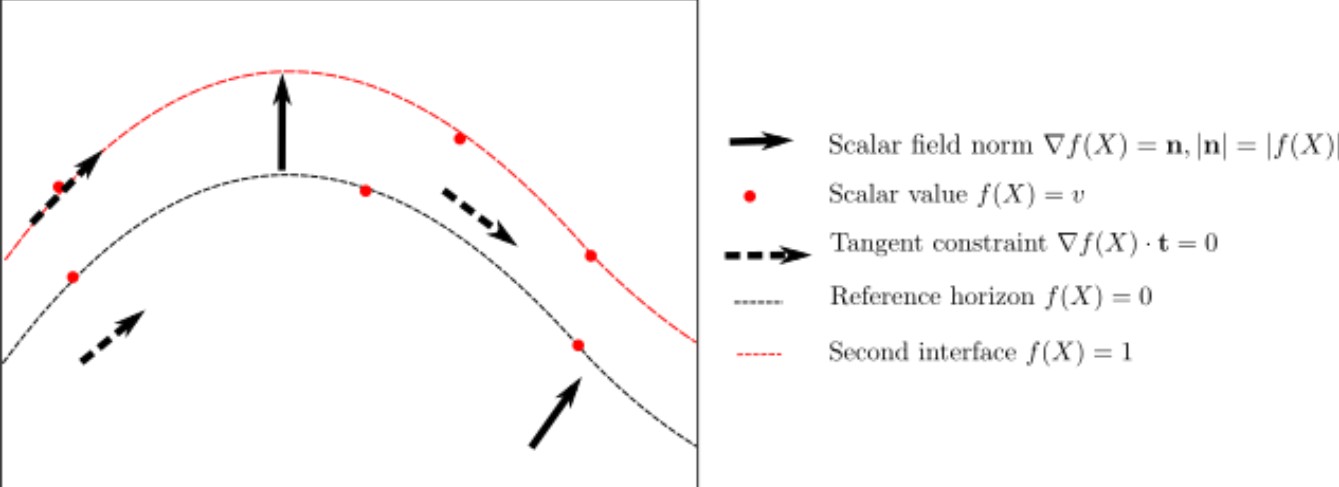

**Figure 1: Schematic showing different types of interpolation constraints that can be applied to an implicit interpolation scheme in 2-D. There are two interfaces the reference horizon with a value of 0 and the next interface with a value of 1. Here we show three types of constraints 1) scalar field norm constraints constrain the orientation of the scalar field and the norm of the implicit function**
**at that location. 2) scalar field value constraints control the value of the scalar field and 3) tangent constraints constraint only the orientation of the implicit function not the norm. Figure adapted from Hillier et al.,** (2014)

When modelling using the potential field approach, the value of the scalar field is inferred through the magnitude of the normal control points. Using the signed distance approach, the value of the scalar field is defined by the value of the observations and effectively controls the thickness of the layers. Orientation constraints either control a component of the orientation e.g.

specifying that the function should be orthogonal to the observation or constrain the magnitude and direction of the norm of the implicit function

All geological observations constrain a component of the implicit function at a location in the model.

- Observations for the location of the geological feature will constrain the value of the scalar field $f(x, y, z) = v$
- Observations for the orientation of the contact can either:

o   constrain the partial derivatives of the function $\nabla f(x, y, z) = \boldsymbol{n}$

o   constrain a vector which is orthogonal to the contact $\nabla f(x, y, z) \cdot \boldsymbol{t} = 0$

It is worth noting that when constraining the partial derivative of the scalar field, the norm of the vector defines the norm of the implicit function which controls the distance between isosurfaces. The sign of the vector must be consistent with the polarity of the structural observation, e.g. for bedding this must be the younging direction. Structural orientations can also be

incorporated into the model using two tangent constraints where $t_1 \times t_2 = n$. In the following sections we will outline the theoretical background for the piecewise linear interpolation, finite difference interpolation and data supported interpolation. Within all approaches, the observations are incorporated by adding observations as constraints into a linear system of equations.





### 2.1.2. Piece-wise linear interpolation

The volumetric scalar field is defined by a piece-wise linear function on a volumetric tetrahedral mesh. In *LoopStructural* the

volumetric tetrahedral mesh creation is simplified by subdividing a regular cartesian grid into a tetrahedral mesh where one

cubic element is divided into 5 tetrahedra.

The linear tetrahedron is the basis of the piecewise linear interpolation algorithm, where the property within the tetrahedron is

interpolated using a linear function:

$$\phi(x, y, z) = a + bx + cy + dz$$

This can be expressed by the values at the nodes (0-3):

$$\phi_0 = a + bx_0 + cy_0 + dz_0$$
$$\phi_1 = a + bx_1 + cy_1 + dz_1$$
$$\phi_2 = a + bx_2 + cy_2 + dz_2$$

$$\phi_3 = a + bx_3 + cy_3 + dz_3$$

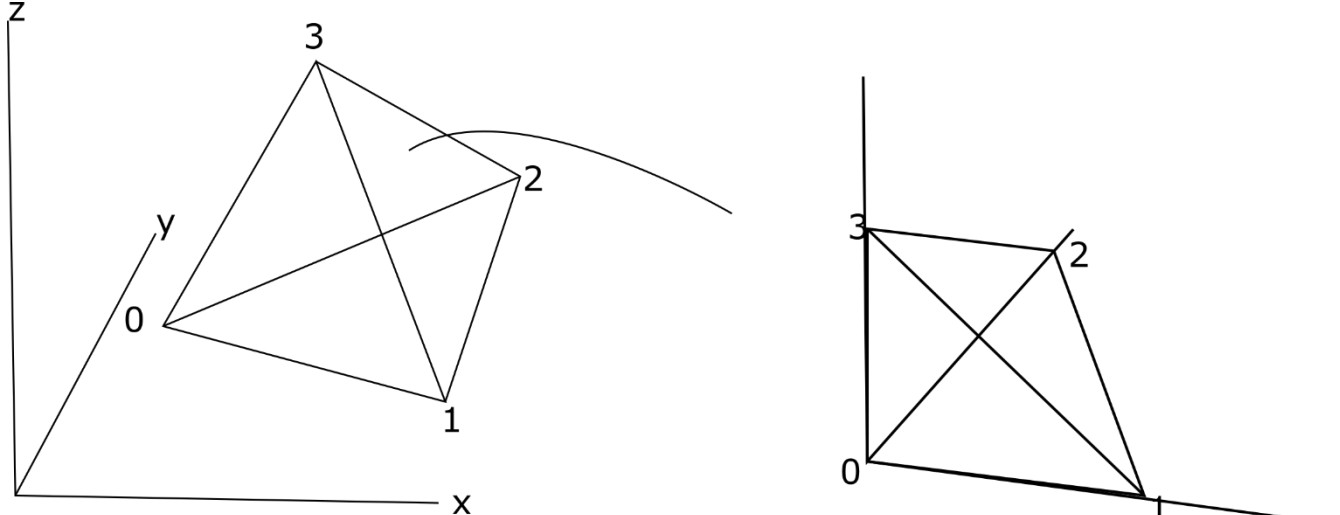

**Figure 2: Schematic diagram showing transformation from tetrahedron in Cartesian space to reference tetrahedron in natural coordinates. This transformation allows for the shape functions and derivatives to be simplified.**

Solving this set of linear equations for $a, b, c, d$ depends on the location of the tetraherdon nodes and has to be recalculated

for every tetrahedron. This can be simplified by applying a coordinate transformation to a reference tetrahedron (Figure 2).

This simplifies the solution and allows for the interpolation to be described by the barycentric coordinates $(c_0, c_1, c_2, c_3)$ of the

tetrahedron. The barycentric coordinates can be used as a local coordinate system for the tetrahedron $(\xi, \eta, \zeta)$:

Since;

$$c_0 + c_1 + c_2 + c_3 = 1$$





$$\xi = c_1$$
$$\eta = c_2$$
$$\zeta = c_3$$

The property within the tetrahedron can be interpolated using the four shape functions below:

$$N_0(\xi,\eta,\zeta) = 1 - \xi - \eta - \zeta$$
$$N_1(\xi,\eta,\zeta) = \xi$$
$$N_2(\xi,\eta,\zeta) = \eta$$
$$N_3(\xi,\eta,\zeta) = \zeta$$


The gradient of the function within the tetrahedron $\partial\phi_T$ can be found by applying the chain rule between the derivative of the shape function within the barycentric coordinates and the partial derivatives with respect to the natural coordinates and Cartesian coordinates:

$$\frac{\partial\varphi_T}{\partial x} = \sum_{i=0}^{3} f(x_i,y_i,z_i)\left(\frac{\partial N_i}{\partial\xi}\frac{\partial\xi}{\partial x} + \frac{\partial N_i}{\partial\eta}\frac{\partial\eta}{\partial x} + \frac{\partial N_i}{\partial\zeta}\frac{\partial\zeta}{\partial x}\right),$$

$$\frac{\partial\varphi_T}{\partial y} = \sum_{i=0}^{3} f(x_i,y_i,z_i)\left(\frac{\partial N_i}{\partial\xi}\frac{\partial\xi}{\partial y} + \frac{\partial N_i}{\partial\eta}\frac{\partial\eta}{\partial y} + \frac{\partial N_i}{\partial\zeta}\frac{\partial\zeta}{\partial y}\right),$$

$$\frac{\partial\varphi_T}{\partial z} = \sum_{i=0}^{3} f(x_i,y_i,z_i)\left(\frac{\partial N_i}{\partial\xi}\frac{\partial\xi}{\partial z} + \frac{\partial N_i}{\partial\eta}\frac{\partial\eta}{\partial z} + \frac{\partial N_i}{\partial\zeta}\frac{\partial\zeta}{\partial z}\right),$$

We use constant gradient regularisation (Caumon et al., 2013; Frank et al., 2007; Mallet, 1992) where the change in gradient
of the implicit function is minimised between tetrahedron with a shared face. The constant gradient regularisation is:

$$\nabla\varphi^{T1} \cdot n - \nabla\varphi^{T2} \cdot n = 0$$

Where $\partial\varphi^{T1}$ is the gradient of the first tetrahedron and $\partial\varphi^{T2}$ is the gradient of the second tetrahedron and $n$ is the normal to the shared face.

### 2.1.3. Finite difference interpolation

The second discrete interpolation approach approximates the interpolant using a combination of tri-linear basis functions on a cartesian grid. The basis functions describe the interpolation as a function of the corners of the cell within which the point where the function is to be estimated falls. For example, to evaluate the value of the implicit function at a point $x_i, y_i, z_i$, first the cell $c$ is found. The local coordinates $(\xi,\eta,\zeta)$ are determined by finding the relative location of the point within the cell.






$$N_0 = \frac{1}{8}(1 - \xi)(1 - \eta)(1 - \zeta)$$

$$N_1 = \frac{1}{8}(1 + \xi)(1 - \eta)(1 - \zeta)$$

$$N_2 = \frac{1}{8}(1 + \xi)(1 + \eta)(1 - \zeta)$$

$$N_3 = \frac{1}{8}(1 + \xi)(1 - \eta)(1 + \zeta)$$

$$N_4 = \frac{1}{8}(1 - \xi)(1 - \eta)(1 + \zeta)$$


$$N_5 = \frac{1}{8}(1 + \xi)(1 - \eta)(1 - \zeta)$$

$$N_6 = \frac{1}{8}(1 + \xi)(1 + \eta)(1 + \zeta)$$

$$N_7 = \frac{1}{8}(1 - \xi)(1 + \eta)(1 + \zeta)$$

The derivative of the function can be calculated by applying the chain rule, in the same way as for the linear tetrahedron,
however in this case to all eight shape functions $N_{0\dots7}$. Different regularisation terms can be used for example (Irakarama et
al., 2018) minimises the sum of the second derivatives:

$$\frac{\partial^2}{\partial_{xx}} + \frac{\partial^2}{\partial_{yy}} + \frac{\partial^2}{\partial_{zz}} + 2\frac{\partial^2}{\partial_{xy}} + 2\frac{\partial^2}{\partial_{yz}} + 2\frac{\partial^2}{\partial_{zx}} = 0$$

Alternatively, a partial differential equation such as the bending energy (Renaudeau et al., 2019) or Gaussian curvature could
be used.

### 2.1.4. Solving discrete interpolation

Using either the piecewise linear interpolator or the finite difference interpolator the scalar field is defined by the node values
of the support. These can be found by solving a system of equations with $M$ unknowns $x_1, \dots, x_M$ (Caumon et al., 2013; Frank
et al., 2007; Mallet, 2004). The unknowns can be found by solving the linear system of equations:
interpolation is a is solved by building a series of linear equations;

$$\mathbf{A} \cdot \mathbf{x} = \mathbf{b}$$

Where $\mathbf{A}$ is an MxN sparse matrix containing the linear constraints and $\mathbf{b}$ the right-hand side vector containing the observation
of constraint value. For example, to integrate value observations the row in the interpolation matrix A would contain the shape
parameters for the cell which the point is contained. The right-hand side would be the value of the scalar field.





The interpolation problem is over-constrained, i.e. $N > M$, and can be solved in a least squares sense. The least squares problem can be solved using a number of different algorithms either directly where $A^T \cdot A$ is directly inverted e.g. using Lower and Upper decomposition or using an iterative algorithm such as Conjugate Gradient. Generally, for large problems an iterative approach is recommended because it requires less memory. *LoopStructural* allows for multiple different solvers to be used for the least squares problem. The default solver is the Conjugate Gradient algorithm implemented in *scipy*. To speed up the solver and in some cases improving the stability of the solution we provide the option of adding a small value (the smallest representable float) can be added to the diagonal of the square matrix ($A^T \cdot A$).

### 2.1.5. Data supported interpolation

Another approach for implicit surface modelling is to use basis functions that are located at the same location as data points. This can either be done using Radial Basis Interpolation where the interpolation problem is attempting to approximate the signed distance field:

$$f(x, y, z) = \sum_{i=0}^{N} w_i \cdot \varphi(X) + P(x, y, z)$$

Alternatively, the problem can be represented using dual co-kriging (Calcagno et al., 2008b; De La Varga et al., 2019; Lajaunie et al., 1997). Where the interpolation algorithm estimates the potential field, which estimates incremental differences between the scalar field for different horizons. Using this approach, the system is separated into two parts 1) the orientation observations which are incorporated using the direction and magnitude and; 2) the difference between the potential field for different horizons.

*LoopStructural* uses *SurfE*, a C++ implementation of the generalised radial basis interpolation (Hillier et al., 2014) for all data supported interpolations. *SurfE* has three approaches for implicit surface reconstruction 1) signed distance interpolation using radial basis functions, 2) potential field interpolation using dual co-kriging (Lajaunie et al., 1997) and 3) signed distance interpolation using a separate scalar field for each surface. The algorithmic interface between *LoopStructural* and *SurfE* allows the user to access all of the interpolation parameters used by *SurfE*. These include access to more sophisticated solvers, as well as the addition of a smoothing parameter into the interpolation.

### 2.2. Modelling Geological Features

There are three ways that rock packages can structurally interact in a geological model:

1. Stratigraphic contacts - the contact between sedimentary layers
2. Fault contacts
3. Intrusive contacts



These geological interfaces can all be affected by deformational structures such as folds, faults and shear zones. In the following sections we will describe how these different geological features are integrated into 3D modelling workflows by describing how different scalar fields interact, how the structural geology of faults and folds are added into the implicit surface description.


### 2.2.1. Stratigraphic contacts

In an implicit geological model, the distribution of stratigraphic packages is defined by the values of a volumetric scalar field. The scalar field is defined by an implicit function that is fitted to observations (location and orientation) defining the geometry of the top or base of a geological unit. A single geological interface can be modelled using a single scalar field, or multiple

conformable interfaces can be modelled using a single scalar field where different isovalues are used to represent the different contacts. A stratigraphic group can be considered as a collection of stratigraphic surfaces that are conformable. When modelling a stratigraphic group, the value of the scalar field represents the distance away from the base of a group of conformable layers.

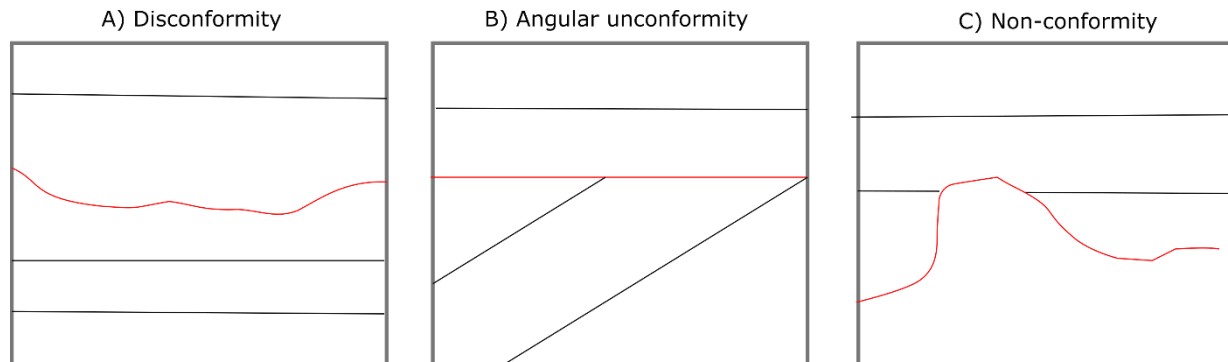


**Figure 3: Unconformities interfaces (red lines) and geological interfaces (black lines) represent a break in depositional history. There are different possible geometries that an unconformity can have A. Disconformity contact between two stratigraphic packages that share a similar geometry. B. An angular unconformity where the younger stratigraphic package defines the geometry of the unconformity C. A nonconformity where the older stratigraphic package defines the geometry of the unconformity.**

An unconformity is a geological interface where the rock units on either side are of significantly different ages usually representing a period of erosion. In Figure 3 the three conventional types of unconformity are shown. In Figure 3A. the unconformity between the units is a disconformity and the geometry of the disconformity is not associated with either stratigraphic packages. The disconformity is usually identified by the significant gap between the age of the rocks. In this type of contact the layers actually share a similar geometry and for the purpose of 3D modelling the units could be represented by

a single stratigraphic group. Angular unconformities (Figure 3B) are observed when erosion occurs after some deformation (the older beds are not horizontal anymore) and before the next deposition of sedimentary layers. As the name suggests the angular unconformity represents a boundary between two differently oriented stratigraphic sequences. In a 3D model an




angular unconformity can be introduced by setting the boundary between the two sequences to be the base of the younger package. In practice, this means that the two groups are modelled with two separate scalar fields. In Figure 3C. a nonconformity

is show, in this type of unconformity the geometry of the older unit defines the base of the younger unit. This could occur when a stratigraphic package is deposited on top of a crystalline basement.

### 2.2.2. Structural frames

A structural frame (Figure 4) is a local coordinate system that is built around the major structural elements of a geological event. In *LoopStructural* structural frames are used for characterising the geometry of folds where the major structural element

is the fold axial foliation (Figure 4B) and the structural direction is roughly the fold axis. A fault frame is a structural frame where the major structural feature is the fault surface, the structural is the fault slip and the intermediate direction is the fault extent (Figure 4). In *LoopStructural*, structural frames are built by first building the major structural feature which typically will have more observations e.g. fault surface location or axial foliations. The structural direction is then built using any available observations of the structural direction e.g. local observations of the fault slip or the fold axis, combined with an

additional constraint which sets the gradient of the scalar field to be orthogonal to the major structural feature. The $3^{rd}$ coordinate, can be built with an arbitrary value constraint, or value constraints to specify the extent of the field in this direction (e.g. for faults -1 and 1 specify the edges of the fault). This value constraint is combined with two global orthogonality constraints specifying that the scalar field should be orthogonal to both the major structural feature and the structural direction.

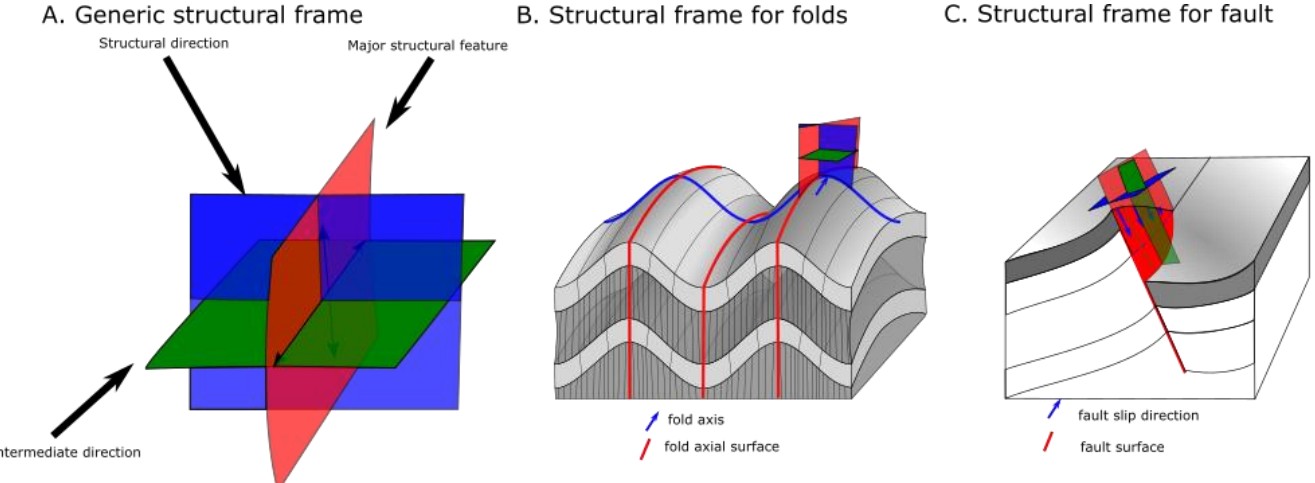


Figure 4: A. Generic structural frame showing isosurfaces for three coordinates. B. Structural frame for characterising a fold. C. Structural frame for characterising fault geometry





### 2.2.3. Faults

*"A fault is a tabular volume of rock consisting of a central slip surface or core, formed by an intense shearing , and a*
*surrounding volume of rock that has been affected by more gentle brittle deformation spatially and genetically related to the*
*fault"* (Fossen, 2010)

When adding faults there are two aspects to modelling the fault: 1) building the fault surface geometry and; 2) integrating the
fault displacement into older surfaces. Where possible, measurements of faults include the movement direction and, the
magnitude of displacement. There are three broad approaches for integrating faults into the implicit modelling framework: 1)
add the fault into the implicit description of the surface (Calcagno et al., 2008a; De La Varga et al., 2019);  2) apply the fault
after interpolating a continuous surface (Godefroy et al., 2018a; Laurent et al., 2013) and 3) represent the foot wall and hanging
wall by separate implicit functions. Regardless of the approach used, the geometry of the fault surface is defined before defining
the geometry of the surfaces displaced by the fault. The fault surface can be interpolated by building a scalar field where the
fault surface is represented by an isovalue.

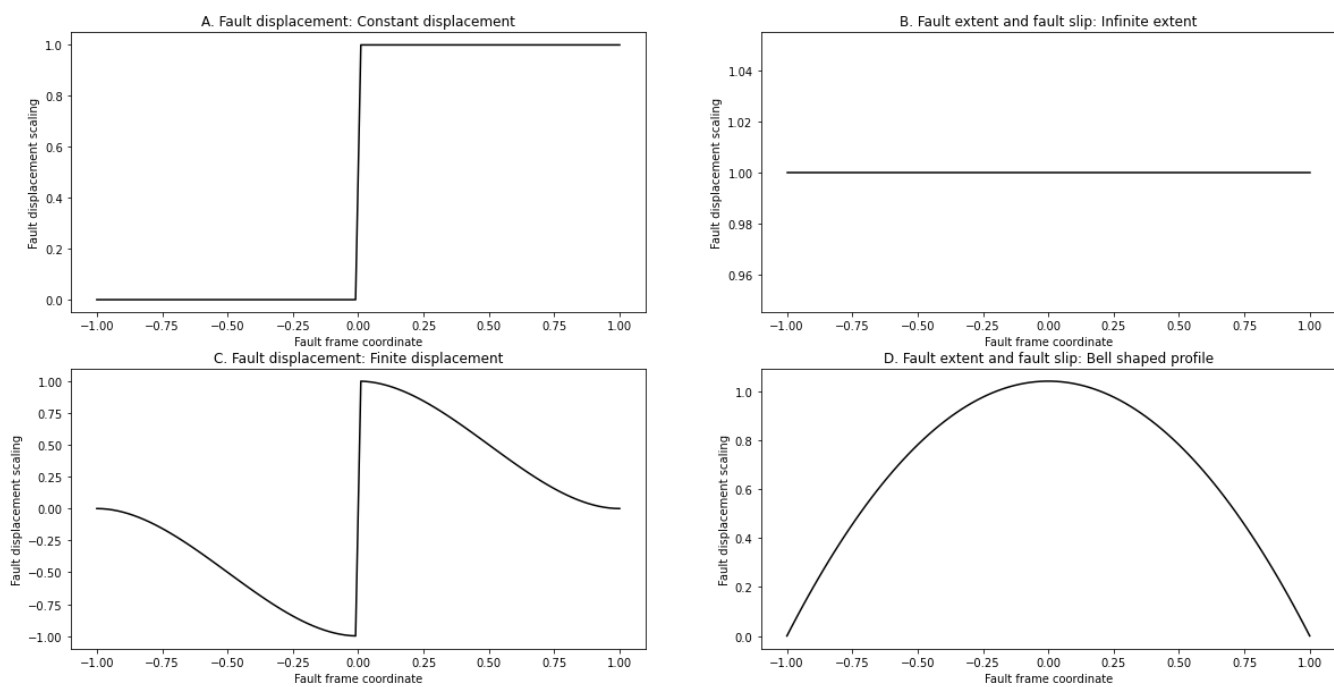


**Figure 5: Fault displacement profiles A. constant displacement profile. B. infinite-extent fault displacement showing no change in
fault displacement along the fault extent or in the slip direction. .C finite-extent fault displacement showing fault displacement
decreasing with distance away from the fault. D) finite-extent fault bell shaped profile for characterising fault displacement along**
**the fault extent or in the slip direction.**

In *LoopStructural* there are two ways of representing faults: 1) the fault kinematics are added into the implicit description of
the scalar field of the faults and applied to the affected scalar field(s) (Grose et al., in prep) and 2) faults are treated as domain





boundaries and separate scalar fields are used to model the hanging wall and footwall of the fault. The kinematics of the fault
are added into the implicit description of the faulted surface. To do this a fault frame is built (Figure 4C) where three

coordinates are interpolated 1) a scalar field representing the distance to the fault surface, 2) a scalar field representing the
distance along the slip direction of the fault and 3) a scalar field representing the extent of the fault. These coordinates can
then be used to define the fault ellipsoid which is a volumetric representation of the area deformed by the fault. The
displacement of the fault can be defined relative to this coordinate system e.g. the displacement of the fault should decay away
from the fault centre along the fault extent and along the direction of displacement using the bell-shaped curve (Figure 5D.).

The displacement may decrease with distance away from the fault centre perpendicular to the fault surface, this can be defined
using the profile in Figure 5C. If the displacement is constant within the model area the curves in Figure 5A and B can be
substituted for C and D respectively. The displacement curves shown in Figure 5 can be substituted for any function of the
fault frame coordinates. The same approach for combining the fault profiles (Figure 5) within the fault frame has been used to
define a volumetric fault displacement field ( Jessell & Valenta, 1996; Godefroy et al., 2018b), the latter of which was adapted

from (Laurent et al., 2013):

$$\delta(x) = D_0\big(f_{0(X)}\big) \cdot D_1\big(f_1(X)\big) \cdot D_2\big(f_2(X)\big)$$

Where $D_{0,1,2}$ are 1-D curves (e.g. any of the curves in Figure 5) describing the displacement of the fault within the fault frame.

The fault frame can be easily built using a discrete implicit modelling approach as additional constraints can be added into the

interpolation to enforce the orthogonality of the three coordinate systems. This is added into the interpolation matrix by adding
a constraint for every element in the mesh where $\nabla\phi_0(x, y, z) \cdot \nabla\phi_1(x, y, z) = 0$. This constraint can be added twice so that
when modelling $\phi_2$ both $\phi_0$ and $\phi_1$ are orthogonal. In general, this means that if the fault orientation, fault trace and fault slip
direction are known, the fault can be modelled. Where the fault slip is unknown, this can be substituted by conceptual
knowledge e.g. enforcing strike slip faults or reverse faults.


### 2.2.4. Folds

Folds are challenging to model using classical implicit interpolation algorithms, because by definition a folded surface has a
symmetry only defined by their axial surface. The symmetry is hard to reproduce by only interpolating orientations of the
folded foliation as this would require to sample orientations in a symmetrical way across the axial surface (Laurent et al., 2016;

Lisle et al., 2007; Mynatt et al., 2007). The regularisation of implicit algorithms are usually defined to minimise some sort of
curvature between observations such as constant gradient regularisation, minimising second derivatives using finite differences
or the weighted combination of infinite basis functions (Calcagno et al., 2008a; Cowan et al., 2003; Frank et al., 2007; Jessell
et al., 2014; Lajaunie et al., 1997; Laurent, 2016; Mallet, 2014). As a result, to model folded geometries the geologist is required



to add interpretive constraints such as synthetic bore holes, cross sections or simply synthetic constraints to produce model
geometries that fit the geologists conceptual idea of the fold (Caumon et al., 2003; Jessell et al., 2014, 2010).

There have been a number of different approaches to incorporating folds into implicit modelling including incorporating the
fold axial surfaces (Laurent et al., 2016; Maxelon et al., 2009), the fold axis (Hillier et al., 2014; Laurent et al., 2016; Massiot
and Caumon, 2010), both these structural elements and fold overprinting relationships (Laurent et al., 2016).

*LoopStructural* implements the following fold constraints: the fold axis, fold axial surface and overprinting relationships
(Laurent et al., 2016) by adding additional constraints into a discrete interpolation approach. A fold frame (Figure 4B and
Figure 6) is built where the principal axes of the fold frame correspond with the direction of the finite strain ellipsoid. The fold
frame allows for the geometry of the folded surface to be defined.

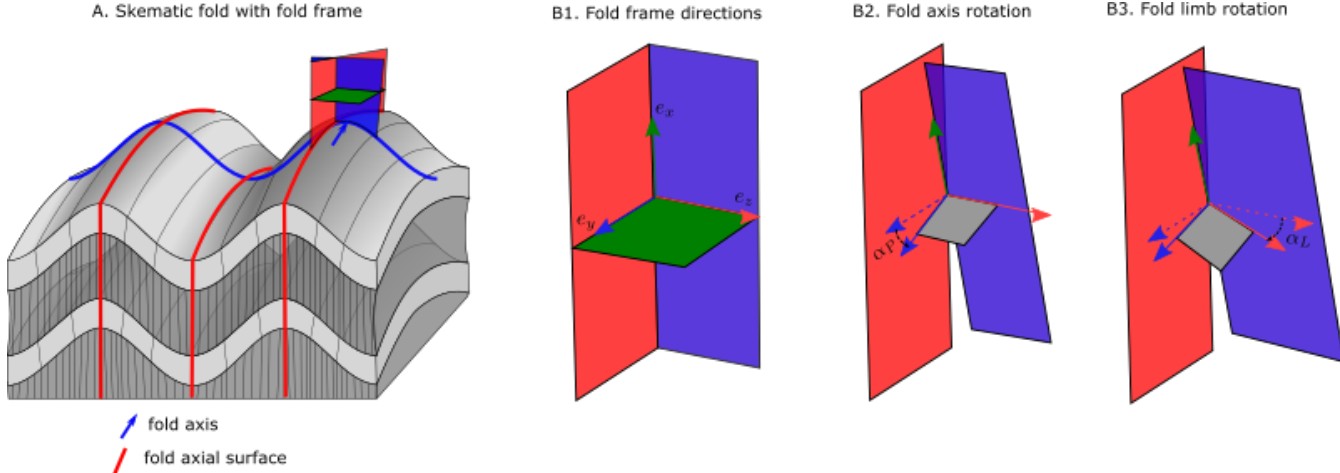

**Figure 6: Schematic diagram of a fold adapted from Laurent et al., (2016) showing: A. fold frame B1. Fold frame direction vectors. B2 fold axis defined by fold axis rotation angle. B3. folded foliation defined by fold limb rotation around fold axis.**

The orientation of a folded surface can be defined within the fold frame by rotating the fold axis direction field by the fold axis
rotation angle (Figure 6B2). The fold direction is defined by rotating the normal to the axial foliation around the fold axis by
the fold limb rotation angle. The orientation of the folded surface is the plane defined by the fold axis vector and the fold
direction vector (Figure 6B3). The fold constraints have been implemented into the piecewise linear interpolator using four
main constraints, where $\varphi(x,y,z)$ represents the implicit function, $\nabla$ represents the gradient, $t$ represents a tetrahedron where
the constraint is applied or $t1$ and $t2$ two tetrahedrons that share a face:

1. fold axis constraint - the folded surface should contain the orientation of the fold axis. $e_y^t \cdot \alpha_P^t \cdot \nabla\varphi(x,y,z) = 0$
2. the folded surface will contain the fold direction (solid red arrow in Figure 6B3) vector. $e_z^t \cdot \alpha_L^t \cdot \nabla\varphi(x,y,z) = 0$
3. the regularisation should only occur within the intermediate structural direction ($e_x$) $e_{x0}^t \cdot \nabla\varphi_{t1}(x,y,z) - e_{x1}^t \cdot \nabla\varphi_{t2}(x,y,z) = 0$



4. A similar fold constraint $e_x^t \cdot \nabla \varphi(x, y, z) = \frac{1}{h_s}$

The fold constraints require two angles to be known throughout the model: the fold axis rotation angle ($\alpha_P$) and the fold limb roation angle ($\alpha_L$). The fold axis rotation angle ($\alpha_P$) is the angle between the fold axis and $e_y$ (Grose et al., 2017).

The fold limb rotation angle ($\alpha_L$) is the angle that defines the orientation of the folded foliation relative to fold axial foliation and will be 0 in the hinge of the fold and positive and negative in the limbs(Grose et al., 2017). Grose et al., (2017) used the fold frame to calculate these angles for observations and then applied interpolation using either Radial Basis Functions, or by

fitting an objective function (a Fourier series) to the rotation angles within the fold frame coordinates. The wavelength of the fold can be estimated by calculating an experimental semi-variogram of the fold rotation angle in the fold frame coordinates. For periodic folding, the experimental variogram has a periodic curve where the first peak indicates the half wavelength of the fold (Grose et al., 2017).

In *LoopStructural* the default approach for fitting the fold rotation angle is to fit a Fourier series using the *scipy.optimize.curve_fit*. The fold axis rotation angle is calculated first, the wavelength is estimated automatically using the gradient descent method on the fold axis of the experimental variogram. The fold rotation angle is estimated by fitting a Fourier series to the observations, the estimated wavelength and Fourier coefficients of $c_0 = 0, c_1 = 0, c_2 = 0$ are used to fit the Fourier series. The fold axis can then be defined throughout the model by applying the rotation of $e_y \cdot R_p$. If the fold axis is constant

(cylindrical folding), a constant fold axis vector can be used.

The fold limb rotation angle is calculated by finding the complementary angle between the normal to the folded foliation and $e_z$ in the plane perpendicular to the fold axis. The fold limb rotation angle can be interpolated by fitting a Fourier series to the observations in the same way as fitting the fold axis rotation angle.


Grose et al., (2018) and Grose et al., (2019) use inverse problem theory to fit a forward model of the fold geometry to the observed fold rotation angles. The joint posterior distribution of the fold parameters (Fourier series coefficients, fold wavelength and a misfit parameter) are sampled using Bayesian inference. This allows multiple fold geometries to be explored without perturbing the datasets. *LoopStructural* does not provide a direct probabilistic interface, however, it is possible to

define a probabilistic representation of the fold geometry curves and add this into the modelling workflow. An example using the python library *emcee* (Foreman-Mackey et al., 2013) is provided in the *LoopStructural* documentation.





## 3. Implementation in LoopStructural

### 3.1. Loop structural design

*LoopStructural* is written using Python 3.6+, using *numpy* data structures and operations. The design of *LoopStructural* follows an object-oriented architecture with multiple levels of inheritance. There are 5 submodules that can be imported into a python environment:

1. **core** - contains the core modelling functionalities and the management of the geological concepts
2. **interpolation** - contains the various interpolation code and supports used to build scalar fields
3. **datasets** - test and reference data sets
4. **utils** - miscellaneous functions
5. **visualisation** - model visualisation tools

The creation and management of different geological objects is managed by the **GeologicalModel**. To initialise an instance, the required arguments are the minimum and maximum extents of the bounding box, which are specified by two separate vectors. The default behaviour is to define a rescaling coefficient as:

$$scale = \max(x_{max} - x_{min}, y_{max} - y_{min}, z_{max} - z_{min})$$

Adding different geological objects can be done through using an instance of **GeologicalModel**. There are four different types of observations that can be incorporated into an interpolation algorithm:

1. *value* - constrain the value of the scalar field at a particular location and can either represent the location of a surface or the distance away from the surface
2. *gradient* - constrains only the gradient of the scalar field e.g. the normal to the scalar field should be orthogonal to two vectors within the gradient plane
3. *tangent* - the scalar field should be orthogonal to a vector
4. *norm* - constrains the direction and magnitude of the scalar field norm

The data can be associated with the **GeologicalModel** using the *set_data(data)* method where *data* is a *pandas* dataframe. When added into the model the data points are transformed into the model coordinate system.

### 3.2. Adding geological objects

Geological objects such as stratigraphy, faults, folding event, and unconformities are all represented by a **GeologicalFeature** or a combination of **GeologicalFeatures**. A **GeologicalFeature** can be evaluated for the value of the scalar field and for the gradient of the scalar field at a location.





There are different ways a **GeologicalFeature** can be added to a **GeologicalModel** depending on the type of geological object that is being modelled. The following functions can be used for adding different geological objects to the **GeologicalModel**.

- *create_and_add_foliation(feature_name, \*\*kwargs)*
- *create_and_add_fault(feature_name,displacement,\*\*kwargs)*
- *create_and_add_fold_frame(feature_name, \*\*kwargs)*
- *create_and_add_folded_fold_frame(feature_name, fold_frame=None,\*\*kwargs)* - adds a fold frame but uses the fold interpolator to model the main structural frame coordinate.
- *create_and_add_folded_foliation(feature_name, fold_frame=None,\*\*kwargs)*
- *create_and_add_unconformity(feature_name,value=0,\*\*kwargs)*
- *create_and_add_domain_fault(feature_name,value,\*\*kwargs)*


All of these functions involve building a volumetric scalar field using an implicit interpolation algorithm. *LoopStructural* allows for different interpolation algorithms to be specified for different **GeologicalFeature**s within the same model. The interpolation algorithm and any parameter definitions are specified by adding additional keyword arguments to the function. Table 1 outlines the possible arguments that can be specified for the interpolator.


### 3.3. Model Output

*LoopStructural* includes a number of helper functions for evaluating the **GeologicalModel** on an array of coordinates within the model. The following functions can be called from a **GeologicalModel** as shown in the code below.

- To evaluate the lithology value at a location the function *evaluate_model(xyz)* returns a *numpy* array containing the
integer ID of the stratigraphy that was specified in the stratigraphic column.
- To evaluate the value of a **GeologicalFeature** at a location within the model the function *evaluate_feature_value(feature_name,xyz)* returns the value of the scalar field that represents the geological feature.
- To evaluate the gradient of a **GeologicalFeature** the *evaluate_feature_gradient(feature_name,xyz)* can be called


Triangulated surfaces can be extracted from a **GeologicalFeature** within *LoopStructural* and exported into common mesh formats e.g. Visualisation ToolKit (.vtk) or Wavefront (.obj). These surfaces can then be imported into external software e.g. ParaView[5].

---

[5] https://www.paraview.org/



### 3.4. Model visualisation

*LoopStructural* has three different visualisation tools that can be accessed from the *LoopStructural.visualisation* module:

1. **LavaVuModelViewer -** LavaVu (Kaluza et al., 2020) is a visualisation module that provides interactive visualisation. We use LavaVu for visualising triangulated surfaces representing the geological interfaces as well as the scalar field representing the implicit function. The creation and manipulation of LavaVu objects is wrapped by the LavaVuModelViewer class which provides an interface to the **GeologicalModel,** Interactive (and static) 3D
visualisation using *LavaVu*.

2. **MapView -** 2D visualisation (cross section, map) using *matplotlib* (Hunter, 2007) that can create a geological map from the resulting geological model. Input datasets can be plotted drawing the location of contacts and the orientation of the contacts using the strike and dip symbology. The scalar field can be evaluated on the map surface, contours can be drawn or the geological model can be plotted onto the map.

3. **FoldRotationAnglePlotter** – Visualisation module for producing S-Plot and S-Variogram plots for a folded geological feature. Plotting is handled using *matplotlib.*

### 4. Examples

### 4.1. Implicit surface modelling

In the first example we will demonstrate modelling two synthetic surfaces using the same scalar field within a model volume
of $(-0.1, -0.1, -0.1)$ and $(5.1,5.1,5.1)$. The observations are two sets of points. The first set forms a surface at points on a regular grid for $z = 4$ and the second set forms a surface at $z = 0$ with noise introduced randomly between 0 and 1.

In Figure 7A. the data points are shown, in B, C and D the same surfaces are interpolated using the three default interpolation algorithms in LoopStructural (PLI, FDI and SurfE). The results for the interpolation using PLI and FDI are very similar, as
both interpolation algorithms use least squares to fit observations whilst minimising a global regularisation term that effectively minimises the second derivative of the implicit function. This means that the interpolant balances fitting the observations with minimising the roughness of the resulting surfaces. The radial basis interpolation used by SurfE is a direct interpolation approach, which means that the interpolant must fit all of the observations (although a smoothing constraint can be used). In this example, because the surfaces are over-constrained to a highly variable pointset the resulting surface is non-manifold
(cannot be unfolded into a flat plane). While this does not necessarily mean the surface is incorrect it is geologically unlikely.





**Figure 7: Comparison of interpolation methods for synthetic surfaces where two isosurfaces are shown coloured by the local z coordinate: A. Input data. B. Surfaces interpolated using PLI. C. Surfaces interpolated using FDI. D. Surfaces interpolated using SurfE, note the lower isosurface has a non-manifold geometry.**

The weighting of the regularisation constraint generally has the biggest impact on the resulting geometry when using the discrete interpolation approaches. In Figure 8 the regularisation constraint is varied from 0.1 (rougher surface) to 1.5 (smoother surface). Lower regularisation constraints result in surfaces that more closely fit the observations at the cost of a more irregular surface. However, even for the lowest regularisation constraints the surfaces still do not fit every observation. There is no

explicit rule for choosing the relative weighting of the regularisation as often it is dependent on the surfaces being modelled. For example, when modelling a surface where the underlying process causing the variation in the data points is non-stationary, a higher regularisation constraint is appealing as the goal of the modelling is to reproduce the effect of this process. However, if the perturbations are the result of a process we are trying to model (probably a stationary process) then a lower regularisation

constraint would be appealing. A smoothing constraint can be added into the radial basis interpolation which aims to increase the smoothness of the resulting surface. The smoothing constraint for data supported methods adds a buffer to how closely the function must fit the observations. In Figure 8 increasing regularisation results in smoother surfaces however with this approach the fit to both surfaces is impacted which can be seen by the change in colour of the surface which represents the local height of the surface.

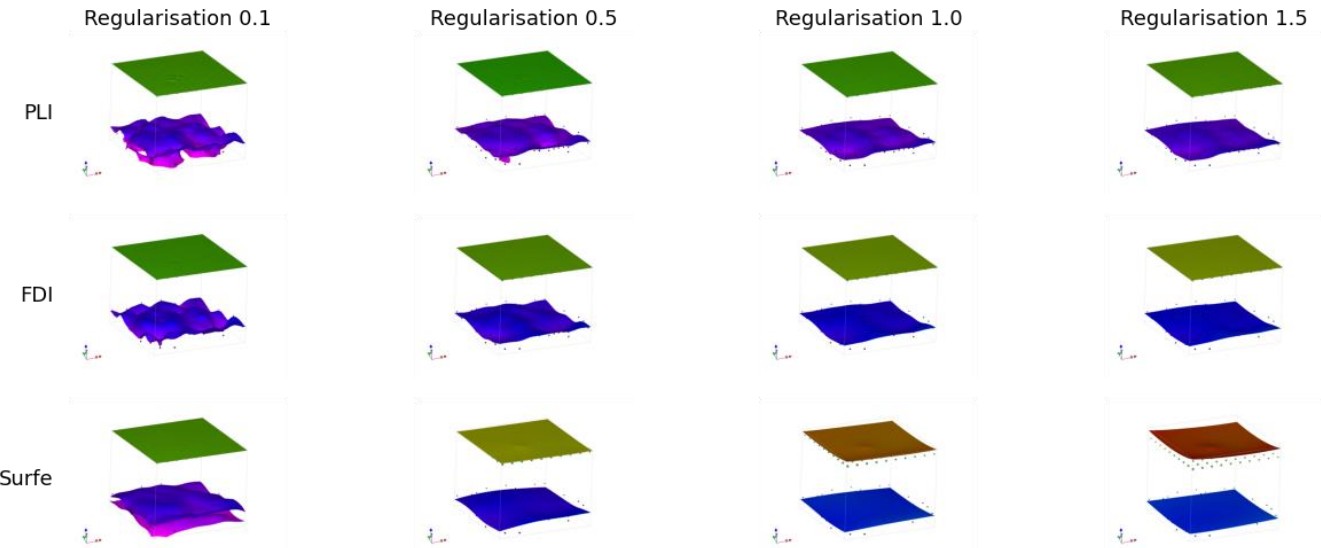


**Figure 8: Implicit surfaces calculated for regularisation constraints (0.1,0.5,1,1.5) using Piecewise Linear Interpolator (PLI),Finite Difference Interpolator (FDI) and radial basis function (SurfE).**

### 4.2. Modelling folds: type 3 interference

To demonstrate the time aware approach for modelling folds we reuse the case study from Laurent et al., (2016). The reference model was generated using Noddy (Jessell and Valenta, 1996) with two folding events forming a type 3 interference pattern:

1. F1 large scale recumbent folding (wavelength: 608 m, amplitude: 435 m, fold axis: N000E/45∘).
2. F2: upright open folding (wavelength: 400 m, amplitude: 30 m)

The structural observations were sampled from a synthetic topographical horizon from three outcrop locations. The axial

foliation to F2, S2 is shown in Figure 9A. The observations of S2 are used to interpolate the major structural feature of the F2 fold frame (Figure 12A). The axial foliation of F1, S1 is shown in Figure 9B and the scalar field value of the interpolated S2 is shown on the map. The fold rotation angle for F2 is calculated by finding the angle between the interpolated S2 field and



the folded S1 field and is shown in the S-Plot for Figure 10A. The red curve in Figure 10A is a Fourier series that is automatically fitted to the observations. The wavelength of the fold is estimated by finding the first peak of the S-Variogram

Figure 10B. Fold constraints are added into the interpolation algorithm using this curve to define the geometry of the fold looking down plunge and the average intersection lineation between the S1 foliation and the interpolated S2 field is a proxy for the fold axis. The interpolated scalar field is shown in Figure 12B. The observations of S0 are shown in Figure 9C and the scalar value of the S1 field is shown on the map. The S-Plot for F1 is shown in Figure 11A and shows two opposing fold limbs in the data points. The red curve shows the Fourier series that characterises the geometry of the fold looking down plunge and

indicates that there are two unobserved fold hinges away from the data points. These constraints are added into the implicit model and the scalar field is shown in Figure 12C.

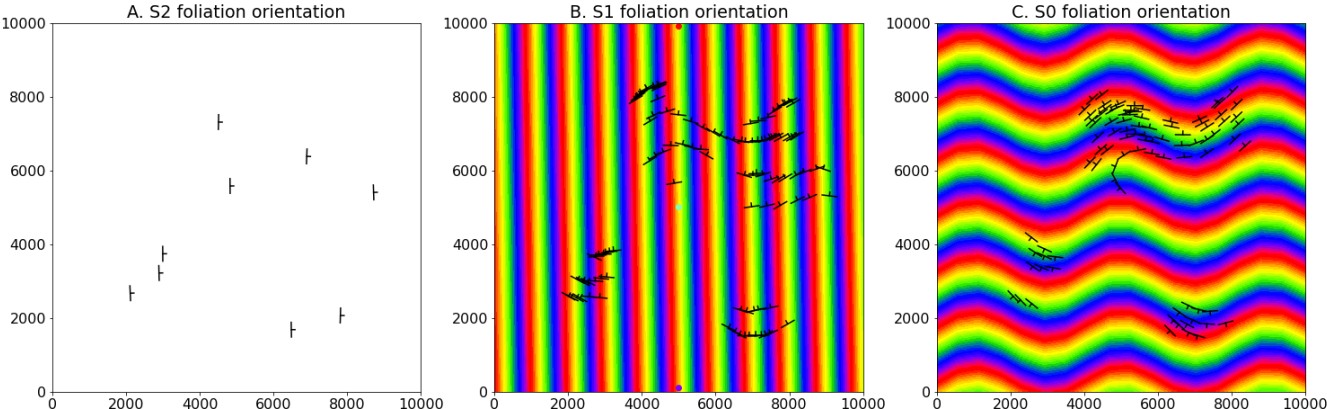

**Figure 9: Structural data for refolded fold. A. observations of S2 B. observations of S1 showing interpolated scalar field of S2 and**

**C. observations of S0 showing interpolated scalar field of S1**

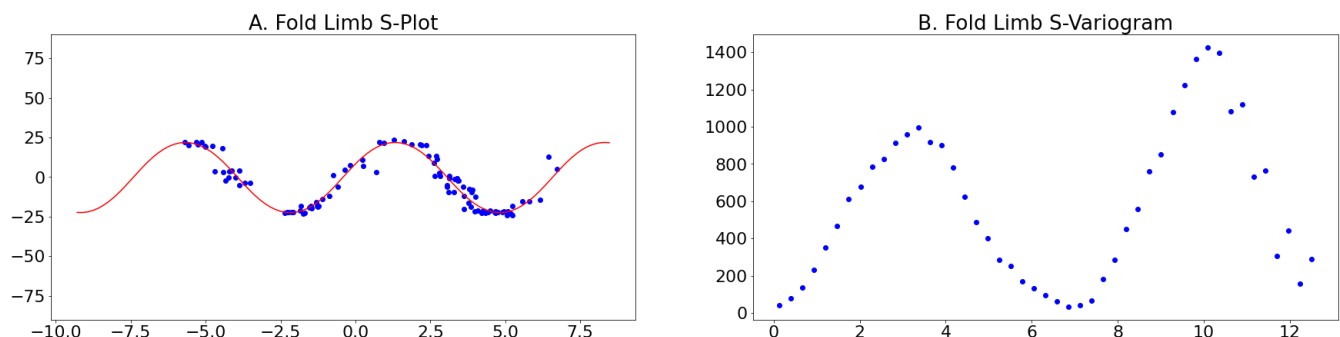

**Figure 10: F2 S-Plot showing the fold rotation angle between observations of S1 and the fold frame S2**





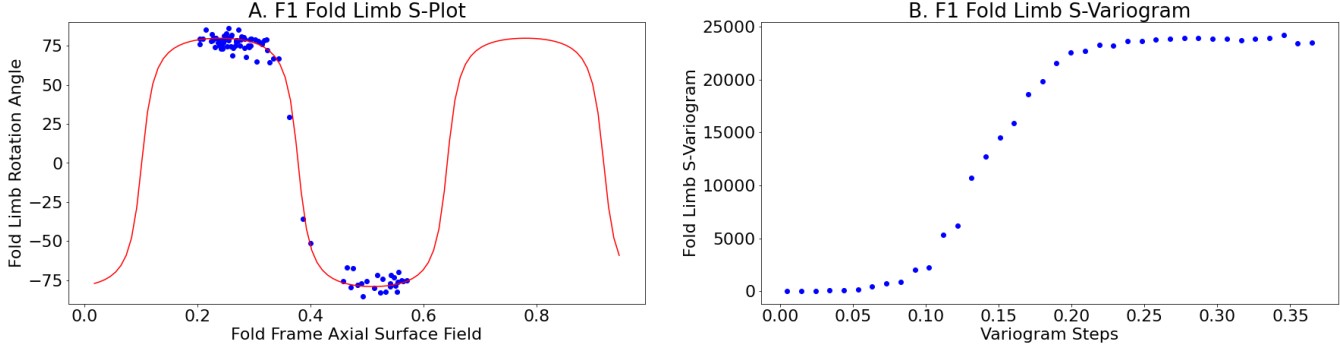


**Figure 11: F1 S-Plot showing the fold rotation angle between observations of S0 and the fold frame S1**

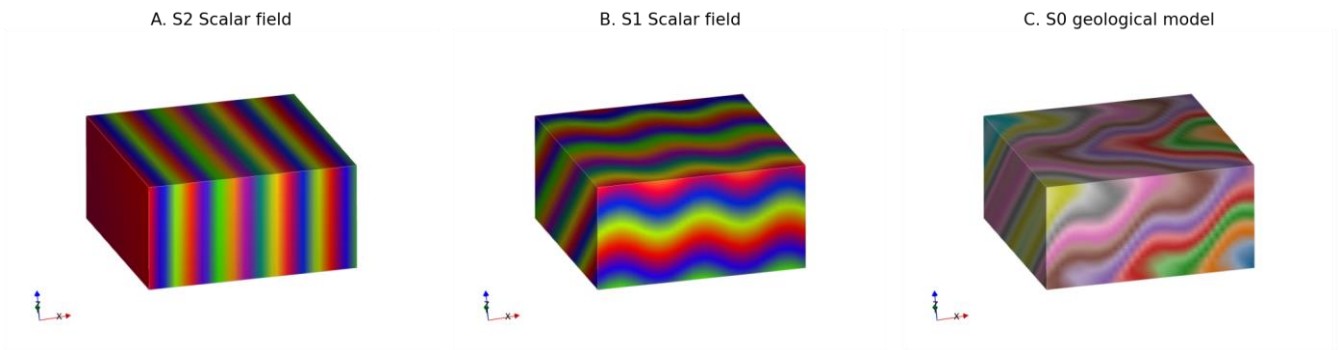

**Figure 12: Scalar fields A. S2, B. S1 and C. bedding**

### 4.3.  Integration with *map2loop*

In the final example we use *map2loop* (Jessell et al., in prep) as a pre-processor to generate an input dataset from regional geological survey maps, the national stratigraphic database and a global digital elevation model. *map2loop* creates a set of augmented data files that can be used to build a geological model in *LoopStructural*. The class method

(*GeologicalModel.from_map2loop_directory(m2l_directory, \*\*kwargs)*) creates an instance of a  **GeologicalModel** from a root *map2loop* output directory.

This example uses a small study area from South Australia using the Geological Survey of South Australia's open access datasets (GSSA, 2020). The model area covers approximately 85 by 53 km within the Finders Ranges in South Australia. The

stratigraphic units within this area are shown in Figure 13A, and the outcropping geology is shown in the geological map (Figure 13B), the patches of the map without any geological units represent shallow Tertiary and Quaternary cover. *map2loop* extracts basal contacts from the outcropping geological units and estimates the layer thicknesses shown in the stratigraphic





column. Within this map area all of the stratigraphic groups share a similar deformation history and area modelled as a single super group. The cumulative thickness is estimated for all of the stratigraphic horizons relative to the Pound Subgroup and is

used to constrain the value of the implicit function. There are 15 faults within the model area with limited geometrical information constraining only the map trace of the faults. As a result, the faults are assumed to be vertical with a vertical slip direction and the displacements are estimated from the geometry on the geological map using *map2loop*. The overprinting relationships of the faults are estimated from the geological map using *map2loop* and are used to constrain the order of the faults in the geological model. The scalar field representing the supergroup is interpolated after the observations of the

stratigraphic horizon (contacts and orientation measurements) are un-faulted using the calculated fault displacements. The modelling workflow is all encapsulated in the (*GeologicalModel.from_map2loop_directory(m2l_directory, \*\*kwargs)*) class method, meaning the geological model can be produced without any user input.

The resulting geological model surfaces are shown in Figure 14 where the surface represents the base of a stratigraphic group and are coloured using the stratigraphic column Figure 13A. The faults in the model are interpolated using a cartesian grid

with 50,000 elements and are interpolated using the finite difference interpolator and the interpolation matrix is solved using the *pyamg* algorithmic multigrid solver (Olson and Schroder, 2018). Stratigraphy is interpolated using a finer mesh with 500,000 elements using the Piecewise Linear Interpolator also using *pyamg*. Using a workstation laptop with an i7 processor and 32gb of RAM the data processing using *map2loop* takes approximately 30 seconds, building the implicit model takes approximately 1 minute and 30 seconds and the rendering of the surfaces on a (100x100x100) cartesian grid takes 3 minutes.

The intersection of the solid geological model and the map surface is shown in Figure 13C allowing for a comparison with the input dataset. The geological model has interpolated the geological packages underneath the surficial deposits.

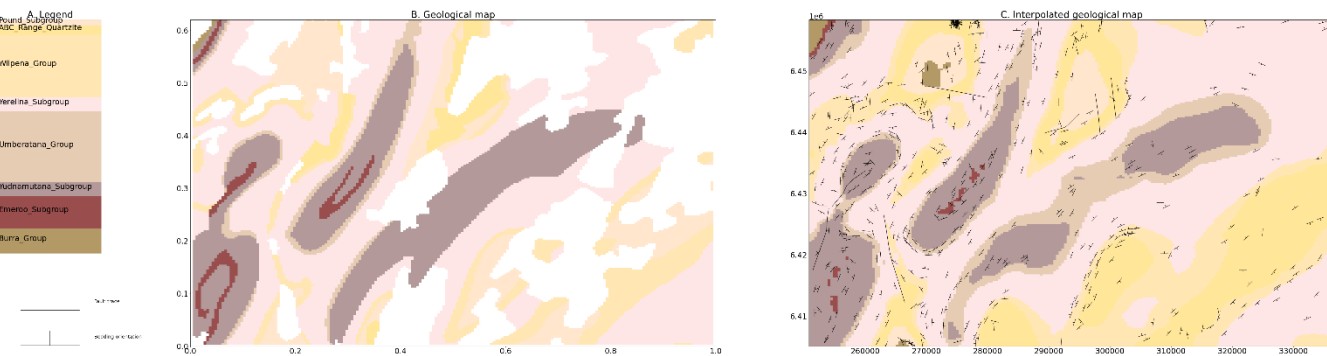

**Figure 13: A. Stratigraphic column for model area showing relative thickness B. Geological map showing bedding and faults C.**
**Geological model shown on map surface**





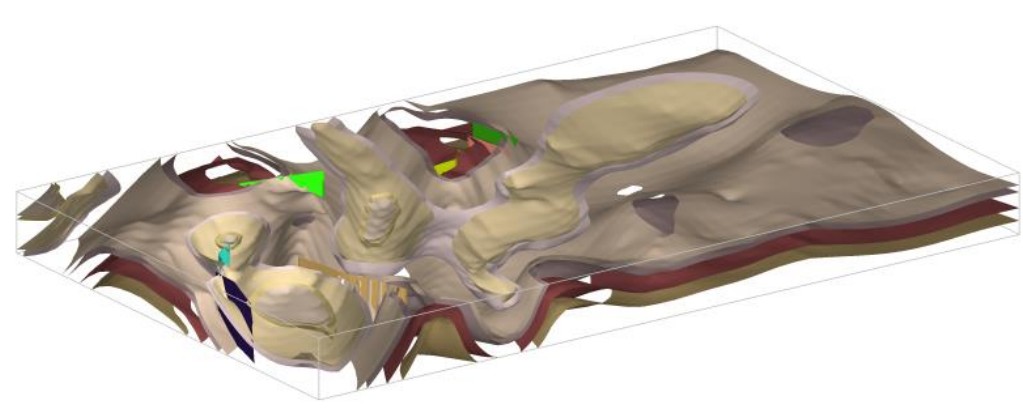

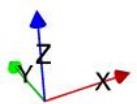

**Figure 14: Geological model from South Australia using map2loop processed data stratigraphic surfaces using colours from Figure 13A and fault surfaces.**

## 5.   Discussion

*LoopStructural* is the 3D geological modelling module for *Loop,* a new open source 3D probabilistic geological and geophysical modelling platform. *LoopStructural* integrates the relative timing of geological features into the description of the

model elements using a time aware modelling approach where the model is built by adding geological features in the reverse order they occur. This is necessary for capturing the complexities of complex structural geometries, for this approach is used for modelling refolded folds Figure 12. In a similar way faults are added backwards in time, this means that the displacements of the faults are applied to the model prior to interpolating the faulted surface. As a result, the fault displacements and overprinting relationships are internally consistent. In comparison, where faults are represented using step functions (Calcagno

et al., 2008b; de la Varga and Wellmann, 2016) the fault displacements are added into the interpolation of the faulted surfaces



using the polynomial trend in the dual cokriging system, meaning the cumulative displacement is determined as the best global fit, rather than incorporating the displacements of individual faults.

*LoopStructural* provides a flexible open source implementation of implicit geological modelling algorithms workflows. The motivation behind developing *LoopStructural* was to create a framework for being able to develop new implicit geological modelling algorithms and tools. *LoopStructural* has native implementation of piecewise linear interpolation (Caumon et al., 2013; Frank et al., 2007; Mallet, 1992, 2004), including the fold constraints (Grose et al., 2017; Laurent et al., 2016) and a finite difference interpolator minimising the second derivative as a regularisation constraint (Irakarama et al., 2018). In Figure 8 we showed that choosing the regularisation weight is somewhat dependent on the quality of the input data set. For this reason, varying the regularisation weight and interpolation approach should be a common step in implicit modelling workflows. The current implementation of the piecewise linear interpolation uses a tetrahedral mesh that is derived from a cartesian grid. A more sophisticated mesh generated from an external mesh generation code could be integrated into *LoopStructural* by overwriting the tetrahedral mesh class with a custom class. Within the *LoopStructural* architecture alternative regularisation constraints could easily be incorporated. For example, it is possible to define custom constraints for implementation within the finite difference scheme, the use simply has to provide a dictionary containing 3D numpy arrays and a relative weighting. New interpolation schemes can be easily implemented using various levels of inheritance to avoid re-writing boiler plate code. For example, the interface with *SurfE* capitalises on the object-oriented design of *LoopStructural*, where the interface between *LoopStructural* and *SurfE* was achieved by creating a new class which inherits the components for the base geological interpolation class. Both the piecewise linear interpolator and finite difference interpolator inherit from a base discrete interpolation class which manages the assembly of the least squares system and the solving of the least squares problem. This object-oriented design allows for the interpolation algorithms to be interchanged and reimplemented without modifying the other aspects of the geological modelling.

A recent focus of 3D modelling research has been to simulate uncertainties by framing the problem as an inverse problem, where the data points are the parameters of the forward model (De La Varga et al., 2019). This allows for additional geological knowledge to be integrated into the model definition such as fault displacement, fault type, fold geometry. Within *LoopStructural*, we have directly integrated many aspects of the geological knowledge into the interpolation schemes and model definition. The fundamental reasoning behind our approach is that the subjective constraints that are required to capture the geological features with standard implicit algorithms will be one of the greatest sources of uncertainty in the model. By incorporating the geological concepts into the geological modelling algorithms, these conceptual uncertainties can be integrated into a probabilistic definition of the geological model. Currently, *LoopStructural* does not have a probabilistic interface however all parameters relating to geological structures (topological ordering, fold geometries, fault displacement and geometries) are accessible from the **model** class functions.





In Figure 14, *map2loop* (Jessell et al., in prep) generates an augmented dataset from the open access geological survey databases (stratigraphic database, DTM, geology shapefiles, structural lines and structural observations). In this example, the total time from data processing to model rendering was approximately 5 minutes. Using discrete implicit modelling means that the complexity of the model is defined by the resolution of the support, rather than the number of observations. Discrete interpolation involves solving the linear equation $A \cdot b = x$ where $A$ is a sparse matrix. Different algorithms can be used for

solving this linear system. For example, the algorithmic multigrid solver used in the Flinders Ranges model can be substituted for the default conjugate gradient solver increasing the interpolation time to 3 minutes. The algebraic solver uses multiple levels of conjugate gradient solvers for coarse grids to approximate the solution to the interpolation problem. The coarse grid solution is then used for improving the solving of the next level. Other approaches to speeding up the linear system could be applied such as using preconditioner for the conjugate gradient solver.


The fault displacement profiles (Figure 5) define the fault displacement within the fault volume, however these conceptual profiles are not fitted to the observations. Godefroy et al., (2018b) interpolates a continuous surface without observations within the fault domain and then use particle swarm optimisation to fit the displacement profiles to the unused observations. *LoopStructural* cannot apply this same approach because all data points are restored (with respect to the fault displacement)

prior to interpolating the faulted surfaces. The displacement estimates calculated by *map2loop* could be used to estimate the displacement profile along the fault trace. The fault displacements could then be optimised using a probabilistic representation of the model geometry parameters. However, defining a specific likelihood function for constraining the fault displacement is challenging and may be specific to the geology in question – e.g. where observations are abundant it would be possible to adopt the technique from Godefroy et al., (2018b) and separate some data from the interpolation, however when dealing with

typical regional scale map sheets most of the observations occur on the surface with limited constraints on the 3D geometry.

## 6.   Conclusions

In this contribution we have introduced *LoopStructural* a new open source Python library for implicit 3D geological modelling. The key features of *LoopStructural* are:

• Implicit 3D geological modelling algorithms using discrete interpolation

   • Implementation of structural geology of folds and faults using structural frames

   • A direct link to *map2loop* for automated 3D geological modelling

   • An object-oriented software design allowing for easy development and extension of the 3D modelling algorithms

*LoopStructural* uses a time aware modelling approach where relative timing between different geological features (folding,

faulting and stratigraphy features) allowing for complex overprinting relationships to be incorporated into the implicit geological models. Folds and faults are encoded using structural frames, a curvilinear coordinate system that is oriented with



the geometry of the major structural feature of the deformational event. Using structural frames, the geometry of folds and faults can be locally characterised similar to how a structural geologist describe the objects in the field.


## Code availability

- LoopStructural is free open-source Python library licensed under the MIT. It is currently hosted on https://github.com/Loop3d/LoopStructural.
- Documentation is available within the package and is hosted on https://loop3d.github.io/LoopStructural
- Jupyter notebooks used for the examples in this paper are available on https://github.com/lachlangrose/loopstructural_paper_examples

## Tables

**Table 1: Interpolation keyword arguments**

| Keyword arguments | Description | Possible values |
|---|---|---|
| Interplator_type | A choice for what interpolator to use e.g. | **'PLI'**, 'FDI', 'Surfe', 'DFI' |
| solver | Which algorithm to solve the least squares problem (for PLI, FDI and DFI) | `cg', `lu', `pyamg', `lsqr', `lsmr',`custom ' |
| nelements | Number of elements in the discrete interpolation approach | |
| buffer | How much bigger to mesh around the model extents | |
| cpw | Weighting of value constraints in discrete least squares problem | |
| gpw | Weighting of gradient constraints in discrete least squares problem | |
| npw | Weighting of norm constraints in discrete least squares problem | |
| tpw | Weighting of tangent constraints in discrete least squares problem | |
| regularisation | Weighting of regularisation constraints in least squares problem | |





| data_region | Buffer around the observations to interpolate scalar field only on a subsection of the mesh |


## Author contribution

LG, GL and LA contributed to the conceptual design of the project. GL wrote an initial version of the fold interpolation code. LG wrote and maintained the code. MJ was involved with the integration with map2loop and ongoing testing of the code. LG prepared the manuscript with contributions from all authors for reviewing and editing.


## Acknowledgements

We acknowledge the support from the ARC-funded Loop: Enabling Stochastic 3D Geological Modelling consortia (LP170100985). Source data provided by GSSA and Geoscience Australia. The work has been supported by the Mineral Exploration Cooperative Research Centre whose activities are funded by the Australian Government's Cooperative Research
Centre Programme. This is MinEx CRC Document 2020/***.

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
