# Peer review of "LoopStructural 1.0: Time aware geological modelling"

_Geoscientific Model Development, 2020_

## Short Comment (SC1) · 21 Dec 2020

Dear authors,

in my role as Executive editor of GMD, I would like to bring to your attention our Editorial version 1.2:

https://www.geosci-model-dev.net/12/2215/2019/

This highlights some requirements of papers published in GMD, which is also available on the GMD website in the 'Manuscript Types' section: http://www.geoscientific-model-development.net/submission/manuscript_types.html

In particular, please note that for your paper, the following requirement has not been

met in the Discussions paper:

- Code must be published on a persistent public archive with a unique identifier for the exact model version described in the paper or uploaded to the supplement, unless this is impossible for reasons beyond the control of authors. All papers must include a section, at the end of the paper, entitled "Code availability". Here, either instructions for obtaining the code, or the reasons why the code is not available should be clearly stated. It is preferred for the code to be uploaded as a supplement or to be made available at a data repository with an associated DOI (digital object identifier) for the exact model version described in the paper. Alternatively, for established models, there may be an existing means of accessing the code through a particular system. In this case, there must exist a means of permanently accessing the precise model version described in the paper. In some cases, authors may prefer to put models on their own website, or to act as a point of contact for obtaining the code. Given the impermanence of websites and email addresses, this is not encouraged, and authors should consider improving the availability with a more permanent arrangement. Making code available through personal websites or via email contact to the authors is not sufficient. After the paper is accepted the model archive should be updated to include a link to the GMD paper.

.

As GitHub is not a persistent archive, please provide a persistent release for the exact source code version used for the publication in this paper. As explained in https://www.geoscientific-model-development.net/about/manuscript_types.html the preferred reference to this release is through the use of a DOI which then can be cited in the paper. For projects in GitHub a DOI for a released code version can easily be created using Zenodo, see https://guides.github.com/activities/citable-code/ for details. Finally note, that according to our new Editorial (v1.2) all data and analysis / plotting

scripts should be made available.

Yours, Astrid Kerkweg
* * *

---

## Referee Comment (RC1) · Italo Goncalves (Referee) · 5 Jan 2021

I would like to congratulate the authors for their effort in developing open-source software for use in the geosciences. I believe we are close to the critical point where open-source tools will see a rise in adoption by the industry, ushering a new cycle of engagement and development that will benefit the whole field.

Overall, the article is clear and well-presented. I consider it in a suitable form for publication, after the minor points below are addressed.

Line 23: It may be interesting to mention the difference between manually drawn explicit surfaces and mathematical explicit surfaces (https://en.wikipedia.org/wiki/Parametric_equation). Some people favor the latter

[Figure]

definition while I personally prefer the former. I don't think there is an "official" definition of geological explicit vs implicit surfaces yet, so this is an opportunity to take a step in that direction.

Line 28: Distance from the surface is not the only way to encode the observations. Gonçalves et al. (2017) work with fixed positive/negative values, while Hillier et al. (2014) use inequality constraints. These would fit in your potential field definition in section 2.1.

Line 119: "black and gray arrows...". Do you mean solid and dashed arrows?

Figure 1: What do you mean by "norm of the implicit function"? If we are dealing with a scalar field, its gradient at a given point has a norm, but I am unfamiliar with the concept of a norm for the field itself. Also, it might be worth mentioning that the gradient constraint is composed of one linear constraint per dimension.

Line 209: Are these alternative regularizers implemented in the package? Do they provide very different results from the standard one? It would be interesting to discuss situations in which one may be preferable over the other, or to point to works that do so.

Line 216 seems to be misplaced.

Line 221: Is the interpolation problem always over-constrained in practice? If I understood correctly, $M$ is the number of nodes and $N$ is proportional to the number of data points. Is that so? Are the regularization constraints added to $N$? If $N > M$, shouldn't the shape of the matrix $\mathbf{A}$ be $N \times M$? Also, it seems that the number of basis functions is defined by $M$ in this case (one per node).

Lines 341-350: A figure illustrating these difficulties would be useful.

Lines 363-383: If the different $\alpha$ values are scalar angles, are they really necessary in the equations, since they represent zero value constraints? It would be useful to write the vectors in boldface, in order to better distinguish them from the scalars. What is

$h_s$?

Line 387: "fold axis of the experimental variogram". Do you mean the experimental variogram of the fold axis rotation angle?

Line 481: Instead of pure noise, perhaps you could sample from a spatially correlated model (this can be easily done through the Cholesky decomposition of the covariance matrix), maybe with 10-20% noise, or even the sum of 2-3 structures with different ranges. It may help to better convey the points made later in the text about prioritization of local/global trends. Exact interpolation of noise will certainly result in unrealistic surfaces. I feel the examples are being somewhat unfair to the RBF model.

Line 484: The acronyms PLI and FDI should be defined right alongside their first mention in the text.

Section 4.2: Being a non-geologist, it is hard for me to visualize the effect the data has on the final model based on the provided figures. Perhaps you could expand Figure 12, showing the measured orientation disks along with some isosurfaces in 3D.

Figure 13: The text is too small.

Line 634: do you mean $\mathbf{Ax} = \mathbf{b}$?

---

## Referee Comment (RC2) · Anonymous Referee #2 · 14 Feb 2021

Grose et al. manuscript is an interesting description of the geological modelling package LoopStructural. The manuscript is fairly well written and the described capabilities of the package are impressive. However, in my opinion the manuscript targets a small audience, who perhaps is already familiar with the package. In that respect, I find the scientific quality of the manuscript fair, and the scientific reproducibility poor, and therefore suggest major revision such that the manuscript becomes more inclusive for a general geological audience.

Below are my major concerns:

1. The mathematics is challenging and in my opinion it does not completely contribute to the understanding of the methods. The interpolation constraints (Fig. 1) are not well explained, tangent and normal constraints seem to be mixed (see eq. for tangent

[Figure]

constraint in Fig. 1, and bullet in line 136), and different terms are not introduced, e.g. t1 and t2 in line 140, N0 to N7 on page 8, phi0, 1 and 2 in line 336, and ex, ey and ez on page 15. I challenge the authors to explain the methods more "conceptually" for a general audience, and leave the mathematics to an appendix section. Certainly, every term in an equation should be explained in the text.

2. Many unclear statements: support (lines 35, 214, 633), principal structural directions (line 86), subdividing a regular cartesian grid into a tetrahedral mesh where one cubic element is divided into 5 tetrahedra (line 145, very difficult to understand), barycentric coordinates (lines 162 and 163), first the cell c is found (line 194, How?), fold axis direction field, fold axis rotation angle, fold direction, fold limb rotation angle (in general this paragraph lines 365-369 is very difficult to understand), a stationary process (line 504), using the polynomial trend in the dual cokriging system (line 596), and containing 3D numpy arrays and a relative weighting (line 610).

3. Many references to "unknown" libraries that a normal user with some knowledge of Python may not know: Theano, emcee, Noddy, LavaVu, map2loop. This phrase is a good example: "The overprinting relationships of the faults are estimated from the geological map using map2loop and are used to constrain the order of the faults in the geological model". How? This seems very challenging and an explanation like this causes more confusion, rather than make things look easier.

4. Key geological concepts, their complexity and implications are not properly acknowledged. Here are specific examples:

a. Unconformities are modelled from a scalar field but it is not clear how variable thicknesses in the case of disconformities and non-conformities are modelled. b. The fault slip vector is often very difficult if not impossible to estimate, yet it defines the fault local structural frame. c. The fold axis is a geometrical construct and not a real element like the fold hinge, yet the fold axis defines the local structural frame for folds. d. How one finds the finite strain ellipsoid on a fold? Doesn't this ellipsoid changes with

the folding mechanism? Is it constant across a fold? Does it make sense to use the strain ellipsoid to define the structural frame? e. The chronology/order of folding and faulting is managed in a very "light" way in examples 2 and 3, yet this often very difficult if not impossible to determine. Would examples 2 and 3 look different if the folding and faulting events are applied with a different order? f. Why are the unit thicknesses of the units in example 3 defined from a single stratigraphic column? Couldn't these thicknesses be variable? Couldn't we use the outcrop trace of the units boundaries to estimate the units' thicknesses? g. Why are the faults in example 3 vertical? Don't the outcrop trace of the faults tell us something about their orientation?;

5. The included examples don't follow a reasonable progression. Example 1 is fine. Example 2 is too complicated and it's not the same example than in the linked note-book. Example 3 is extremely complicated and it seems to be used to illustrate the power of the package to "connect" to other libraries, to do impressive stuff on short time. However, this does not help much the reader. I would rather like to see a much simpler example, perhaps illustrating the challenges mentioned in the last paragraph of the discussion.

6. Section 3 reads like the manual of the package rather than a clear explanation on the design of the program, input, output and visualisation. This section should be modified considerably so that it clearly explains how the program works, without calls to very long "code" sentences, which by the way can be included in an Appendix.

7. For the discussion, I would have liked to see a section about the limitations (and chal-lenges) of the software (every software has limitations). For example: Can LoopStruc-tural model fault-propagation folds? Can it model sediment-growth (growth strata)? What are the challenges?

8. For a presentation paper on a software package, it is rather unfortunate that the package is so difficult to install. I am a Mac user with some knowledge of Python, and I spent 2 hours without success trying to install the package. I used pip, docker,

etc. and I could not do it. This of course prevented me to properly follow up the linked Jupyter notebooks. Very unfortunate. I think the authors should provide much more clear online instructions (possibly videos) about how to install the package.

In summary, I like very much the manuscript and I have no doubt LoopStructural is a great contribution. However (and unfortunately), the manuscript and accompanying online resources target a small audience, probably already familiar with the package and related methods. I hope this review can contribute to a new version that considers a more general audience of geoscientists (not only structural geologists) who could benefit from the use of these methods.

—————————————

---

## Editor Comment (EC1) · Andrew Wickert (Editor) · 6 Mar 2021

Following the comments on the manuscript, both reviewers and myself see value in the work. The senior editor notes compliance issues with GMD standards for open-source software and its connection with the publication, which indeed I should have caught, and which should be straightforward to address. The first reviewer gives minor comments and a positive review, whereas the second reviewer gives a detailed, earnest, and negative review, which indicates significant difficulties understanding the software libraries used and actually installing and testing LoopStructural. Based on the detailed comments of Reviewer 2, I suggest that you revise both the text of your document and the users' manual to ensure that your code is usable.

[Figure]

Following Reviewer 2's remarks, I tried to reproduce their issues on installation. I am using Ubuntu.

- `pip3 install LoopStructural`: Success. Perhaps test Python 2/3 compatibility

- But when I imported Loop: `ModuleNotFoundError:  No module named 'skimage'`

- After apt installing skimage:  `ImportError:  cannot import name 'marching_cubes' from 'skimage.measure'`

At this point, it started to seem to me that there are issues related to dependency versions and to the indicated dependencies employed in your PyPI repository. When ready for publication, such installations should obviously be smooth; I am sure that they are on your machines, but I suggest that you reach out to colleagues to test installation on a range of computers and OS'es while preparing your resubmission.

Best of luck in your improvements, and I look forward to viewing your revised paper and code.

---

## Editor Comment (EC2) · Andrew Wickert (Editor) · 14 Mar 2021

Thank you for your notes on the installation. I updated the PyPI (pip) install and received a new error, which seems to possibly relate to a difference in memory access between Python and C++. I have opened an Issue on GitHub (https://github.com/Loop3D/LoopStructural/issues/58); let us continue over there to try to fix this, as that will streamline any back-and-forth.

---

## Author Comment (AC1) · 14 Mar 2021

Thank you for the comments. We will use Zenodo for the release of LoopStructural and the examples that are associated with the manuscript when it is ready for publication.

The installation issues have been noted and have been addressed on multiple fronts, which should make the process more robust than before.

1) We have now provided a docker container that can be used to run LoopStructural with all of the required dependencies. This is the preferable way of running the code as it ensures that the system LoopStructural is being run on is a clean installation. The docker image is hosted on docker hub as loop3d/loop and can be pulled using docker

pull loop3d/loop.

2) We have precompiled python wheels for unix, mac and windows that can be downloaded/installed using pip into a clean python environment. These wheels are automatically built when a new version of LoopStructural is released.

3) The library can be installed from source, however this can require setting up a C++ compiler. We provide instructions for this for unix on the documentation web page.

It is also possible to view the example notebooks using mybinder and google colab. Although the interactive visualisation is not working on mybinder, it does work on google colab.

We are working on a conda installation from out anaconda channel, loop3d. This is a work in progress and currently works for unix but not windows and has not been tested on osx.

I will post another reply to outline the responses to the other reviewers regarding the contents of the manuscript.

---

## Author Comment (AC2) · 14 Mar 2021

**Response to reviewer two**

Grose et al. manuscript is an interesting description of the geological modelling package LoopStructural. The manuscript is fairly well written and the described capabilities of the package are impressive.

Thank you for the comments and the thorough review of the manuscript.

However, in my opinion the manuscript targets a small audience, who perhaps is already familiar with the package. In that respect, I find the scientific quality of the manuscript fair, and the scientific reproducibility poor, and therefore suggest major revision such that the manuscript becomes more inclusive for a general geological audience.

We have addressed the suggested comments and believe that they will increase the audience of the manuscript. The installation issues have been addressed and the library can be installed on the cloud (google colab). The paper is a model description paper and as a result does include some of the technical aspects of the model, however we have moved the specifics into the appendix to improve readability.

Below are my major concerns:

1. The mathematics is challenging and in my opinion it does not completely contribute to the understanding of the methods.

We have moved the blocks of equations (shape function definitions) to the appendix of the paper as they aren't necessary to the paper but are important for the model description.

The interpolation constraints (Fig. 1) are not well explained, tangent and normal constraints seem to be mixed (see eq. for tangent C1 constraint in Fig. 1, and bullet in line 136), and ex, ey and ez on page 15.

The confusion is due to a mistake, we stated the tangent constraints are orthogonal to the contact and this should have been orthogonal to the gradient of the implicit function or parallel to the contact. We have fixed this. We have also defined the norm constraints explicitly as the partial derivatives to avoid any further confusion

I challenge the authors to explain the methods more "conceptually" for a general audience, and leave the mathematics to an appendix section. Certainly, every term in an equation should be explained in the text.

We have moved some of the maths to an appendix section and ensured that every term is explained.

and different terms are not introduced, e.g. t1 and t2 in line 140

In this case $t1$ and $t2$ were simply referring to two tangent observations as described in text " Structural orientations can also be incorporated into the model using two tangent constraints where $t_1 \times t_2 = n$".

The two tangents can be any pair of vectors that are both orthogonal to the gradient norm (n), usually these are the dip vector and the strike vector. We have modified this section to make it more clear

, N0 to N7 on page 8, phi0, 1 and 2 in line 336,

In the case of the trilinear shape function we define $N_{0\ldots7}$ after describing the shape functions, we have moved the shape functions into an appendix because they are not necessary in the body of the text.

2. Many unclear statements: support (lines 35, 214, 633), principal structural directions (line 86), subdividing a regular cartesian grid into a tetrahedral mesh where one cubic element is divided into 5 tetrahedra (line 145, very difficult to understand), barycentric coordinates (lines 162 and 163), first the cell c is found (line 194, How?), fold axis direction field, fold axis rotation angle, fold direction, fold limb rotation angle (in general this paragraph lines 365-369 is very difficult to understand), a stationary process (line 504), using the polynomial trend in the dual cokriging system (line 596), and containing 3D numpy arrays and a relative weighting (line 610).

We have made some changes to the text for these comments.

- Supports refer to the discrete interpolation support
- Principal structural directions, refer to the orientation of the structure being modelled.
- Subdividing a cartesian grid into a tetrahedral mesh is a computer graphics technique, a single cube can become 5 tetrahedron – the specifics of this are not relevant to the understanding of the method but it is important to state how the tetrahedral mesh was generated.
- Barycentric coordinates are the coordinate system of the tetrahedron
- A cell can be found the same as a pixel in an image. The coordinates of the points are divided by the step vector and the modulus of the results gives the index of the cell.
- The description of the fold constraints was not referred back to the figure showing these elements, this should make it easier to understand.
- Stationary process is a geostatistical term to describe a random process where the samples are all sampled from the same model. We have removed this reference as it necessary to understanding.
- Polynomial trend in the dual co-kriging system refers to the global trend that is modelled within the cokriging system.

3. Many references to "unknown" libraries that a normal user with some knowledge of Python may not know: Theano, emcee, Noddy, LavaVu, map2loop. This phrase is a good example: "The overprinting relationships of the faults are estimated from the geological map using map2loop and are used to constrain the order of the faults in the geological model". How? This seems very challenging and an explanation like this causes more confusion, rather than make things look easier.

All of the libraries except Theano were referenced with associated publications. For example, map2loop has a paper that is also submitted in GMD in the same special issue, we have referenced the paper when first introducing the library in line 73. The map2loop paper provides a complete overview of the map deconstruction process. We have added a statement into this paper to refer the reader to the map2loop paper.

Theano is mentioned describing gempy, another open source 3d modelling package where a reference is included. We have removed the reference to Theano as its not necessary.

These libraries are all used or can be used within loopstructural so need to be referenced but they provide a functionality that is not a direct contribution of LoopStructural therefore should not be included in this paper.

4. Key geological concepts, their complexity and implications are not properly acknowledged. Here are specific examples:

Geological models are an approximation (and simplification) of the geometry of a geological feature. This means that some of the complexities and implications need to be simplified. We would argue that LoopStructural provides more parameters and ability to control these complexities than other packages. We have added a paragraph into the discussion to expand on this topic, but many of the challenges that the reviewer mentions are well known issues in geological modelling but are not within the scope of this paper.

a. Unconformities are modelled from a scalar field but it is not clear how variable thicknesses in the case of disconformities and non-conformities are modelled.

As shown in Figure 3, nonconformities are modelled where the older unit defines the unconformity surface. Disconformities are modelled using a separate scalar field (Section 2.2.1), however this assumes that the geologist has observations defining the disconformity surface which may be challenging but is outside of the scope of this study.

b. The fault slip vector is often very difficult if not impossible to estimate, yet it defines the fault local structural frame.

This is true and we have added shown in the new example how the fault slip vector can be perturbed LoopStructural. If we do not include the fault slip vector in the fault displacement, the kinematics will be incorrect. Adding in a variable that we do not know, means that the modeller can change this to fit their conceptual model and use the geological model as a tool for understanding the geology not simply as a static representation of the map.

c. The fold axis is a geometrical construct and not a real element like the fold hinge, yet the fold axis defines the local structural frame for folds.

The fold axis can be observed in rocks by looking at the intersection lineation between the folded foliation and the axial foliation. For more information about the fold interpolation constraints the papers Laurent et al., 2016 and Grose et al, 2017,2018,2019 provide the justification and background.

d. How one finds the finite strain ellipsoid on a fold? Doesn't this ellipsoid changes with C2 the folding mechanism? Is it constant across a fold? Does it make sense to use the strain ellipsoid to define the structural frame?

We use the directions of the finite strain ellipsoid, not the magnitude of the vectors defining the ellipsoid. The directions of the finite strain ellipsoid are related to the shortening direction, and will be relatively constant across a fold. The foliation that develops in the XY plane of the finite strain ellipsoid can be measured in the field, and provides the starting point for the structural frame. The specifics of the fold modelling have been covered in the previously mentioned papers.

e. The chronology/order of folding and faulting is managed in a very "light" way in examples 2 and 3, yet this often very difficult if not impossible to determine. Would examples 2 and 3 look different if the folding and faulting events are applied with a different order?

The power of loopstructural is that the chronology is encoded in the model if it is not known then it can be changed and the geologist can observe how the model changes geometry. In general, where sufficient data exists for fold geometries the geologist will have a reasonably good control on the

overprinting relationships. See the papers on fold modelling Laurent et al., 2016 and Grose et al, 2017,2018,2019.

f. Why are the unit thicknesses of the units in example 3 defined from a single stratigraphic column? Couldn't these thicknesses be variable? Couldn't we use the outcrop trace of the units boundaries to estimate the units' thicknesses?

The outcrop trace of the unit boundaries is used to estimate the unit thicknesses within the map2loop process. Implicit modelling requires a single scalar value to be assigned to the interface between surfaces. As described in the background section this can either be chosen prior to building the model or can be extracted from the model depending on the constraints used. This does not prevent thickness variation in the resulting units as the magnitude of the gradient norm of the scalar field can vary – as shown in the first case study.

g. Why are the faults in example 3 vertical? Don't the outcrop trace of the faults tell us something about their orientation?;

This comment is probably more appropriate for Jessell et al., 2021, where map2loop is introduced. In general, there are no observations of the fault dip and the topographic relief is limited meaning it is difficult to fit a plane to the fault surface. The least biased estimate is to assume that the faults are vertical with a vertical slip vector. This is a parameter that can be varied to understand how uncertainty propagates through the model as mentioned in the discussion.

5. The included examples don't follow a reasonable progression. Example 1 is fine. Example 2 is too complicated and it's not the same example than in the linked notebook. Example 3 is extremely complicated and it seems to be used to illustrate the power of the package to "connect" to other libraries, to do impressive stuff on short time. However, this does not help much the reader. I would rather like to see a much simpler example, perhaps illustrating the challenges mentioned in the last paragraph of the discussion.

Example 2 has been modified in the notebook to match the documentation, the example demonstrates modelling multiple foliations which is a logical progression from modelling a single foliation.

Example 3 is used to demonstrate the link between map2loop and LoopStructural. For a geologist wanting to make a model of a map area, this link will be invaluable and shows how minimal modelling knowledge is required to build an initial model.

The challenges illustrated in the last paragraph of the discussion involve framing the fault kinematics/geometry as an inverse problem. This is something that could be done using loopstructural as the modelling engine but is another body of work to do.

However, we agree that this topic is interesting and does highlight how useful LoopStructural is. We have added another map2loop case study, to demonstrate how the fault slip vector can be changed in LoopStructural and how this is important for the resulting geometries.

6. Section 3 reads like the manual of the package rather than a clear explanation on the design of the program, input, output and visualisation. This section should be modified considerably so that it clearly explains how the program works, without calls to very long "code" sentences, which by the way can be included in an Appendix.

This section has been modified to highlight the design of loopstructural, the function names have been removed from text and only referenced as the API.

7. For the discussion, I would have liked to see a section about the limitations (and challenges) of the software (every software has limitations). For example: Can LoopStructural model fault-propagation folds? Can it model sediment-growth (growth strata)? What are the challenges?

Loopstructural is a geometrical modelling library and not a mechanical/kinematic modelling library. If the appropriate data is provided to model these structures then LoopStructural will capture them. There is no geometrical forward model for fault propogation folds, or sediment-growth. As shown in the first case study the discrete interpolation approach can capture surfaces where the distance between the surfaces is not constant.

These are both very specific use cases for 3D modelling, and to model them without suffient observations will require more geological knowledge to be integrated into the model. We have done this for folds and faults individually. We have extended the discussion to discuss the limitations of LoopStructural.

8. For a presentation paper on a software package, it is rather unfortunate that the package is so difficult to install. I am a Mac user with some knowledge of Python, and I spent 2 hours without success trying to install the package. I used pip, docker, C3 etc. and I could not do it. This of course prevented me to properly follow up the linked Jupyter notebooks. Very unfortunate. I think the authors should provide much more clear online instructions (possibly videos) about how to install the package.

We acknowledge that the installation process has not been reliable. We have significantly improved the process for installing LoopStructural and have included automated deployment and testing for windows, linux and macos builds.

Using the recent releases, we have also tested using loopstructural in google colab and have added instructions into the documentation for this. This will allow a user to run LoopStructural on the cloud without needing to set up their own environment.

The provided docker file will work if you have a docker environment. Since submitting the paper we have also created some tutorial videos that will be linked on the documentation web page.

If a user still has trouble getting the library to work we suggest adding an issue on the repository.

In summary, I like very much the manuscript and I have no doubt LoopStructural is a great contribution. However (and unfortunately), the manuscript and accompanying online resources target a small audience, probably already familiar with the package and related methods. I hope this review can contribute to a new version that considers a more general audience of geoscientists (not only structural geologists) who could benefit from the use of these methods.

Thank you for the thorough review. We believe that after addressing the comments in this review the paper will be more suitable for people wishing to use LoopStructural.

---

## Author Comment (AC3) · 14 Mar 2021

I would like to congratulate the authors for their effort in developing open-source software for use in the geosciences. I believe we are close to the critical point where open-source tools will see a rise in adoption by the industry, ushering a new cycle of engagement and development that will benefit the whole field. Overall, the article is clear and well-presented. I consider it in a suitable form for publication, after the minor points below are addressed.

Thank you for your positive review.

Line 23: It may be interesting to mention the difference between manually drawn explicit surfaces and mathematical explicit surfaces (https://en.wikipedia.org/wiki/Parametric_equation). Some people favor the latter definition while I personally prefer the former. I don't think there is an "official" definition of geological explicit vs implicit surfaces yet, so this is an opportunity to take a step in that direction.

We have rephrased this section to define the geologist's definition of explicit as manually drawn in contrast to the mathematical definition.

Line 28: Distance from the surface is not the only way to encode the observations. Gonçalves et al. (2017) work with fixed positive/negative values, while Hillier et al. (2014) use inequality constraints. These would fit in your potential field definition in section 2.1.

This is true – we have changed this sentence to show that the distance is only way of encoding the geological observations.

Line 119: "black and gray arrows. . .". Do you mean solid and dashed arrows? Figure 1: What do you mean by "norm of the implicit function"? If we are dealing with a scalar field, its gradient at a given point has a norm, but I am unfamiliar with the concept of a norm for the field itself. Also, it might be worth mentioning that the gradient constraint is composed of one linear constraint per dimension.

This was a typo, it is meant to be the norm of the gradient of the implicit function. We have fix this.

Line 209: Are these alternative regularizers implemented in the package? Do they provide very different results from the standard one? It would be interesting to discuss situations in which one may be preferable over the other, or to point to works that do so.

They haven't been implemented currently but would easy to implement. The comparison between interpolation methods is a topic in itself and is something that will hopefully be possible using LoopStructural. One of the challenges in a comparison is choosing the appropriate case study(s) that does not preferentially favour a particular interpolation scheme. For this reason we have only briefly touched on this topic in the first case study and we hope that LoopStructural will be a suitable platform for this type of comparison study to be performed.

Regarding implementing the different regularisation terms. We have added a sentence in the manuscript to outline how the interpolation can be easily modified without writing boiler plate code.

Line 216 seems to be misplaced.

Yes it was, it has been removed.

Line 221: Is the interpolation problem always over-constrained in practice? If I understood correctly, M is the number of nodes and N is proportional to the number of data points. Is that so? Are the regularization constraints added to N? If N > M, shouldn't the shape of the matrix A be N ×M? Also, it seems that the number of basis functions is defined by M in this case (one per node).

You are correct it should be NxM. Regularisation is added to N. Number of basis functions is defined by the number of elements so not directly M.

Lines 341-350: A figure illustrating these difficulties would be useful.

This topic is covered in detail in Laurent et al., 2016. We have added a reference to this to avoid repetition.

Lines 363-383: If the different α values are scalar angles, are they really necessary in the equations, since they represent zero value constraints? It would be useful to write the vectors in boldface, in order to better distinguish them from the scalars. What is hs?

We have changed this section it was incorrect. The constraints are now defined with respect to the fold component vectors (fold axis and fold direction). These are calculated using the rotation angles specified as described in text.

Hs is the expected thickness of the stratigraphy. It is used to constrain the length of the gradient norm. We have added this into the text.

Line 387: "fold axis of the experimental variogram". Do you mean the experimental variogram of the fold axis rotation angle?

Yes, we have fixed this.

Line 481: Instead of pure noise, perhaps you could sample from a spatially correlated model (this can be easily done through the Cholesky decomposition of the covariance matrix), maybe with 10-20% noise, or even the sum of 2-3 structures with different ranges. It may help to better convey the points made later in the text about prioritization of local/global trends. Exact interpolation of noise will certainly result in unrealistic surfaces. I feel the examples are being somewhat unfair to the RBF model.

We have modified the example to be a combination of a few structures and gaussian noise. The addition of the noise is to highlight the impact of noise on how the interpolator fits. This example is not intended to be unfair to RBF, but a way of highlighting the importance of understanding the interpolation process. Specifically exact interpolation (RBF/cokriging) will work well on clean data, whereas the discrete interpolation will work better on noisy data but may add too much smoothing clean data. As mentioned in the previous comment, a more thorough review of the interpolation schemes is required but is outside of the scope of this study.

Line 484: The acronyms PLI and FDI should be defined right alongside their first mention in the text. Section 4.2: Being a non-geologist, it is hard for me to visualize the effect the data has on the final model based on the provided figures. Perhaps you could expand Figure 12, showing the measured orientation disks along with some isosurfaces in 3D.

PLI/FDI are defined in section 2 now and we have changed the visualisation to orientation disks. We have added html interactive figures for each of the examples in the appendix – it is difficult to communicate the 3D nature of the models without a 3D visualisation environment.

Figure 13: The text is too small.

This has been fixed

Line 634: do you mean Ax = b?

Yes, this has been fixed.

---

## Author Response (AR1)

**Changes for LoopStructural GMD paper**

We have made a number of changes to the paper, the major changes are outlined below.

1. Unnecessary equations have been removed from the paper and put into an appendix section.
2. Example 1 has been modified to use a combination of functions and noise, rather than pure noise
3. Added an example where the fault slip directions have been perturbed
4. Description of loopstructural has been revised to be more conceptual and not include unnecessary references to the code
5. Resulting models for folds and south Australian example are saved as html objects for interactive visualisation
6. Added appendix showing how tetrahedral mesh is generated from cartesian grid

We have also made significant changes to the library, including updating the documentation and installation process.

1. LoopStructural is built on github actions for every release and the python binaries are uploaded automatically to pypi for windows, linux and macosx. This will mean it is easier for users to install LoopStructural.
2. A docker container loop3d/loop has been built including all of the dependencies used to create and run the examples in this paper
3. LoopStructural can be installed and run on google colab (excluding surfe and map2loop)
4. The library and case studies are persistently archived on zenodo.

---

## Author Response (AR2)

Summary of changes – Correction

Thank you for the handling of the manuscript. We have made the following minor changes following your comments.

- The equations showing the normal vector constraints were reverted back to the original description, this is more concise and actually consistent with the notation in the manuscript where the normal vector is referred to as $\boldsymbol{n}$
- References to libraries were highlighted in italics, vectors have been emphasised with bold and a couple of minor typos corrected
- The mathematics was reviewed
- x was replaced with $\times$ for matrix description